



# Climatic variations during the Holocene inferred
# from radiocarbon and stable carbon isotopes in a
# high-alpine cave
Caroline Welte[1,2], Jens Fohlmeister[3,4], Melina Wertnik[1,2], Lukas Wacker[1], Bodo Hattendorf[5], Timothy I.
Eglinton[2], Christoph Spötl[6]
[1] Laboratory of Ion Beam Physics, ETHZ, Otto-Stern Weg 5, 8093 Zurich, Switzerland,
[2] Geological Institute, ETHZ, Sonnegstrasse 5, 8092 Zurich, Switzerland
[3] Potsdam Institute for Climate Impact Research, Telegrafenberg, 14473 Potsdam, Germany
[4] GFZ German Research Centre for Geosciences, Section 'Climate Dynamics and Landscape
Development', 14473 Potsdam, Germany
[5] Laboratory of Inorganic Chemistry, D-CHAB, ETHZ, Vladimir-Prelog Weg 1, 8093 Zurich, Switzerland
[6] Institute of Geology, University of Innsbruck, Innrain 52f, 6020 Innsbruck, Austria
Keywords: radiocarbon, stable carbon isotopes, LA-AMS, speleothems, high-alpine cave
Email:
C. Welte: cwelte@phys.ethz.ch
J. Fohlmeister: jens.fohlmeister@pik-potsdam.de
M. Wertnik: wertnikm@phys.ethz.ch
L. Wacker: wacker@phys.ethz.ch
B. Hattendorf: bodo@inorg.chem.ethz.ch
T. I. Eglinton: timothy.eglinton@erdw.ethz.ch
C. Spötl: christoph.spoetl@uibk.ac.at




## Abstract

A novel technique making use of laser ablation coupled online to accelerator mass spectrometry (LA-AMS) allows analyzing the radiocarbon ($^{14}C$) concentration in carbonate samples continuously at high spatial resolution within very short analysis times. This new technique can provide radiocarbon data similar to the spatial resolution of stable carbon (C) isotope measurements by isotope-ratio mass spectrometry (IRMS) and, thus, can help to interpret $\delta^{13}C$ signatures, which otherwise are difficult to understand due to numerous processes contributing to changes in C-isotope changes ratios. In this work we analyzed $\delta^{13}C$ and $^{14}C$ on the Holocene stalagmite SPA 127 from the high-alpine Spannagel Cave (Austria). Combined stable carbon and radiocarbon profiles allow to identify three growth periods characterized by different $\delta^{13}C$ signatures: (i) the period > 8 ka BP is characterized by relatively low $\delta^{13}C$ values with small variability combined with a comparably high radiocarbon reservoir effect (expressed as dead carbon fraction, dcf) of around 60%. This points towards C contributions of host rock dissolution and/or from an old organic matter (OM) reservoir in the karst potentially mobilized due to the warm climatic conditions of the early Holocene. (ii) Between $3.8 - 8$ ka BP a strong variability in $\delta^{13}C$ reaching values from -8 to +1‰ with a generally lower dcf was observed. The $\delta^{13}C$ variability is most likely caused by changes in gas exchange processes in the cave, which are induced by reduced drip rates as derived from reduced stalagmite growth rates. Additionally, the lower dcf indicates that the OM reservoir is contributing less to stalagmite growth in this period possibly as a result of reduced precipitation or because it is exhausted. (iii) In the youngest section between $2.4 - 3.8$ ka BP, comparably stable and low $\delta^{13}C$ values combined with an increasing dcf reaching up to 50% are again hinting towards a contribution of an aged OM reservoir in the karst.

## Introduction

Understanding the climate of the past is the key for understanding how climate and environment will change in the future. Insights into paleoclimate are gained through the study of archives with stalagmites being a prominent example for a terrestrial archive. Stalagmites can grow continuously over thousands to tens of thousands of years (Fairchild et al., 2006). Caves hosting stalagmites are present on all continents (with the possible exception of Antarctica) and uranium-series disequilibrium dating allows to build robust chronologies (Richards and Dorale, 2003; Scholz and Hoffmann, 2008). Trace-element and isotope data of stalagmites allow the reconstruction of climatic conditions in the past. For example, the oxygen isotope composition ($\delta^{18}O$) is generally interpreted as a combination of a temperature and a precipitation signal. The interpretation of the stable carbon isotope signature ($\delta^{13}C$), however, is more challenging since additional local effects, such as vegetation changes (e.g., (Bar-Matthews et al., 1999; Denniston et al., 2007)), the carbonate dissolution mechanism (e.g., Fohlmeister et al. (2010), Lechleitner et al. (2016)) and in-cave fractionation processes (e.g., Mattey et



al. (2016); Spötl et al. (2005)) may have an influence and little is known about the relative magnitude of these processes. Besides the stable C isotopes, radiocarbon ($^{14}$C) decaying with a half-life of ~5700 yrs (Kutschera, 2013) can be a valuable tool in speleothem research (e.g., (Bajo et al., 2017; Lechleitner et al., 2016)). So far this isotope has not been fully exploited in speleothem sciences, mostly due to the time-consuming sampling and processing as well as the comparably high costs associated with the analyses.

In most karst systems, dissolution of the carbonate host rock is driven by soil-derived carbonic acid. In this case, the two major soil-derived C sources contributing to the $\delta^{13}$C values of the speleothem are pedogenic $CO_2$ from the degradation of soil organic matter (SOM) and root respiration that acidifies the meteoric water as it percolates through the soil and the karst rock. Recently, evidence was found for an additional C source stemming from $CO_2$ derived from the oxidation of old organic matter (OM) in the deep vadose zone (Bergel et al., 2017; Noronha et al., 2015). If a radiocarbon independent chronology for the stalagmite exists, the dead carbon fraction (dcf) can be derived through comparison of the measured $^{14}$C profile in the stalagmite ($F^{14}C_{stal}$) with the $^{14}$C atmosphere's signature ($F^{14}C_{atm}$) of the same time Genty and Massault (1997):

$$dcf = \left(1 - \frac{F^{14}C_{stal}}{F^{14}C_{atm}}\right) \cdot 100\% \qquad (1)$$

Values can range from a few % up to 70% (Bajo et al., 2017; Southon et al., 2012) and commonly vary within a single speleothem with time. The magnitude of the dcf is influenced by multiple factors, such as the age of soil OM, contributing to soil gas $CO_2$ production (Fohlmeister et al., 2011b) and consequently altering the $^{14}$C concentration in the stalagmite. Also the $CO_2$ partial pressure (p$CO_2$) in the soil plays an important role, with a complex relationship between the amount of soil gas p$CO_2$ and the dcf (Fohlmeister et al., 2011b). Additionally, the conditions of karst dissolution, i.e. open vs. closed system (Fohlmeister et al., 2011a; Hendy, 1971), affect the dcf. In a more open system, the dcf is low because the percolating water can continuously exchange C with the soil gas $CO_2$ leading to a $^{14}$C concentration in the stalagmite that is dominated by the near-atmospheric soil $^{14}$C signature. Fractionation and gas exchange processes in the cave are also potential candidates for modulation of the dcf. These main factors driving the dcf in turn are influenced by numerous parameters such as hydrological and environmental conditions above the cave. Several studies (e.g., (Griffiths et al., 2012); Noronha et al. (2014)) showed that during periods of increased rainfall the dcf is enhanced, which is explained by either accelerated SOM decomposition rates and the resulting increased mean age of soil gas $CO_2$ or, more likely, by a shift towards more closed-system conditions under higher precipitation regimes.



An increasing number of cave systems are reported where carbonate dissolution occurs even if no
significant soil layers exists above the cave. Acidic conditions in the seepage water are achieved via
oxidation of pyrite or other sulfide minerals disseminated in the bedrock (Bajo et al., 2017; Lauritzen,
2001; Spötl et al., 2016). In this case the C isotope composition in the drip water is dominated by the
bedrock, and the dcf is therefore expected to be relatively high (>50%). Under those conditions the
$\delta^{13}$C values of the speleothems reflect those levels of the bedrock, i.e. are shifted closer towards 0‰
compared to more depleted $\delta^{13}$C values of speleothem $CaCO_3$ of around -12 and -10‰, for cave
systems with a soil and vegetation cover. An overview of relevant processes as well as the resulting dcf
and $\delta^{13}$C are summarized in Table 1.
*Table 1 Summary of expected $\delta^{13}$C and dcf values in stalagmite $CaCO_3$ for different dominant processes. Note, that in natural*
*samples often a mixture of these processes with different proportions contributed to the stalagmite, which complicates the*
*interpretation.*

| Process | | Expected* $\delta^{13}$C (‰) | Expected* dcf (%) |
|---|---|---|---|
| **Carbonate dissolution via carbonic acid** | open-system | < -10‰ | Comparably low, i.e. around 10% |
| | closed-system | > -10‰ | Comparably high, i.e. close to 50% |
| **Carbonate dissolution via oxidation of pyrite** | | Close to 0‰ | Very high, i.e. > 50% |
| **Old OM contribution to seepage water acidification** | | Shift to more negative values | Shift towards higher values (> 50% possible) |


The stalagmite under study grew in the Spannagel cave (Tyrol, Austria), a high-alpine cave system that
was investigated in many studies in the context of palaeoclimate and palaeoenvironmental research
(e.g., (Fohlmeister et al., 2013; Spötl and Mangini, 2010)). The gneiss covering the cave-bearing marble
contains interspersed fine-crystalline pyrite and is topped by a thin soil layer with sparse vegetation. It
was hypothesized that the oxidation of pyrite contributes considerably to the dissolution of the host-
rock marble and, hence, to the growth of stalagmites and flowstones, in particular during cold climate
periods when there is no soil present at this high altitude (Spötl and Mangini, 2007). During some
interglacials including the Holocene, when alpine soils are present in the catchment of the cave´s drip
water, sulfide oxidation and soil-derived $CO_2$ may operate in tandem. Consequently, the stable C
isotope signal of stalagmites from this cave is expected to be complex.
The aim of this study is to gain deeper insights into the C dynamics in this cave by highly spatially
resolved in-situ $^{14}$C analyses of a Holocene stalagmite. This study takes advantage of a recently
introduced method of laser ablation coupled to accelerator mass spectrometry (LA-AMS; Welte et al.
(2016a); Welte et al. (2016b)), which reaches a similar spatial resolution as micro-milling for stable





isotope analysis, e.g., Spötl and Mattey (2006). Using the combined $^{14}$C and $\delta^{13}$C records as well as the
$\delta^{18}$O signal (Fohlmeister et al., 2013), we explore the key processes influencing the carbon isotope
composition of speleothems in this cave and gain a better understanding of the potential and limits of
$^{14}$C analysis of carbonates using LA-AMS.

## Materials & Methods

1. Sample
Spannagel cave is located in the Tux Valley (Zillertal Alps, western Austria) and opens at 2531 m above
sea level. It forms a more than 12 km-long system of galleries and short shafts, which developed in a
Jurassic marble tectonically overlain by gneiss. This superposition does not only allow for high-
precision U-series dating of stalagmites (due to their relatively high U contents), but also gives rise to
carbonate dissolution via sulfuric acid stemming from pyrite oxidation. The thin alpine soil coverage
provides an additional pedogenic source of acidity and the interplay between the two processes is
reflected by highly variable stable C isotope values as well as dcf in Spannagel speleothems. Stalagmite
SPA 127 was found in situ in the eastern part of the cave system, which was never ice-covered during
the Holocene. The stalagmite grew from 8.45 to 2.24 ka BP with an average growth rate of
approximately 25 µm/a as confirmed by nine U/Th-ages (Fohlmeister et al., 2013).
The 15 cm-long polished slab of the stalagmite analyzed in this study was first used for stable isotope
analysis where sampling was performed along the extension axis. For LA-AMS analysis, the same
section was used but broken in two pieces at a distance from top (dft) of approximately 10 cm, which
will be referred to as "top piece" and "bottom piece" (compare Fig. 1).

2. Stable isotope analysis
Subsamples for stable carbon isotope analysis were micromilled at 100 µm increments and measured
using an automated online carbonate preparation system linked to a triple collector gas source mass
spectrometer (Delta$^{plus}$XL, ThermoFisher, Bremen, Germany) at the University of Innsbruck. Values are
reported relative to the Vienna Pee Dee Belemnite standard. The long-term precision of the $\delta^{13}$C values
(1σ standard deviation of replicate analyses) is 0.06% (Spötl, 2011). The respective $\delta^{18}$O values have
been published earlier (Fohlmeister et al., 2013).

3. Radiocarbon analysis using LA-AMS
By focusing a laser on the surface of a solid sample at sufficiently high energy densities, a small portion
of material is ablated and can be used for trace element or isotopic analysis allowing for fast and
spatially resolved analysis (Gray, 1985; Koch and Guenther, 2011). $^{14}$C analysis of SPA 127 was



performed by LA coupled with AMS (Welte et al., 2016a; Welte et al., 2017). For this study, a slightly
modified LA-AMS setup was used reaching a smaller spot size (75x140 µm$^2$) and higher energy
densities of up to 8 J/cm$^2$ allowing for increased signal intensities, i.e. $^{12}$C-currents. With LA-AMS a
quasi-continuous data stream is produced at 10 sec intervals in the AMS. This is the minimal
integration time of the AMS and together with the laser spot width d and the scanning velocity v
defines the spatial resolution R according to R = d + v · 10 sec.
LA-scans were placed as close as possible to the stable isotope tracks in order to facilitate matching
between the two data sets (Fig. 1). However, the tracks are further away from the central stalagmite
axis, where layers are often curved, resulting in a potential offset between stable isotope and
radiocarbon data of up to several hundred micrometers, with the outer LA-scan appearing somewhat
older than the stable isotope record. On the "top piece" of SPA 127 two subsequent scans in opposite
direction were performed, first from young to old (T1) and then vice versa (T2) on the same track with
a scanning velocity of 20 µm/s and a laser energy density of approximately 5 J/cm$^2$. On the "bottom
piece" a total of three analyses were performed: the initial scan from old to young (B1: 10 µm/s, 1-
2 J/cm$^2$) was followed by a second repeated scan from bottom to top (B2: old to young, 25 µm/s,
8 J/cm$^2$) after removing the top ~0.5 mm of the sample surface by mechanical polishing. The second
scan was necessary to ensure that the unusual $^{14}$C signature observed in the oldest part of the
stalagmite during the first scan (see section "Results") was not the result of a potentially contaminated
surface. A final third analysis (B3) consisting of two scans performed in opposite directions was
performed at 20 µm/s and 5 J/cm$^2$. Processing of the raw $^{14}$C data was performed using in-house
standards also analyzed by LA-AMS for blank subtraction and standard normalization (marble, F$^{14}$C = 0
and coral standard, F$^{14}$C = 0.9445 ± 0.0018). A Savitzky-Golay (SG) filter is applied to the recorded $^{14}$C
signal of B3, which is a smoothing method that reduces noise while maintaining the shape and height
of peaks (Savitzky and Golay, 1964). In brief, a polynomial is fitted to a sub-set of the data points and
evaluated at the center of the approximation interval. Two parameters, namely the number of points
defining the approximation interval and the maximum polynomial order, can be defined. The
smoothing has been applied to the two sub-scans of B3 (from "old to young" and vice versa) as well as
to the combined data to ensure robustness of the filter. Corresponding uncertainties are estimated
from the square root of the sum of the squared difference between the measured F$^{14}$C value and the
SG fit at each point within the interval. This value is then divided by the square root of the difference
between the interval length (number of data points) and the maximum order allowed for the
polynomial, which is equivalent to the degree of freedom.



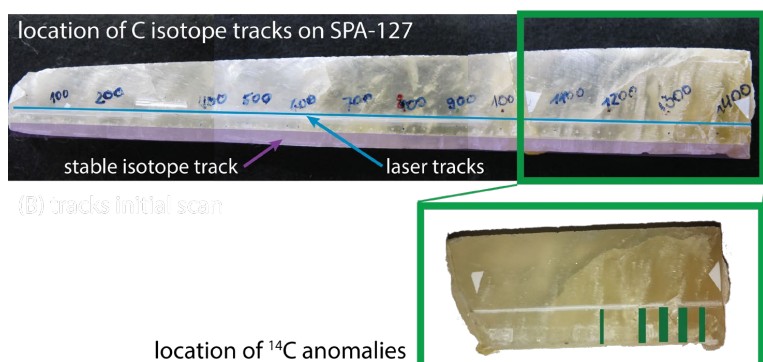


Fig. 1 Picture of SPA 127 (top is left). Top and bottom piece (green box indicates bottom slab) with location of the stable
isotope track marked in purple and LA-AMS test track in blue (the tracks corresponding to the data presented in this work
were placed next to the test tracks). The total length of the slab is 14.6 cm. The green box shows the bottom piece after
removal of approximately 0.5 mm from the sample surface with the repeated LA-AMS laser track and approximate locations
(green areas) of layers exhibiting anomalies during LA-AMS.

192

### 4. **Fourier Transform Infrared spectroscopy (FTIR) analysis**

Fourier Transform Infrared spectroscopy (FTIR) is a standard non-invasive technique for material
analysis (Derrick et al., 2000). The coupling of the IR spectrometer with a microscope enables micro-
analysis, while the development of focal plane array (FPA) detectors allowed to upgrade point
measurements to imaging. Attenuated total reflection (ATR) is a sampling technique, which requires
no sample preparation other than a flat surface and is independent of the thickness of the sample as
it is performed upon contact of the sample with the ATR crystal, which is a medium of high refractive
index.

The analysis of the speleothem was performed on a Spotlight 400 FT-IR Imaging System (Perkin Elmer,
Massachusetts, USA), equipped with a MCT array (mercury cadmium telluride) detector. The
stalagmite was placed under the microscope on a motorized stage and focused to measure the
spectrum between 4000 and 520 cm$^{-1}$ using a micro-ATR objective germanium crystal at 4 cm$^{-1}$
spectral resolution and 32 scans. The detection area of a measurement was 300 x 200 μm$^2$ in diameter
(spot size). An area of 1.9 x 3.4 mm in the vicinity of the abnormal $^{14}$C/$^{12}$C behavior observed by LA-
AMS was selected for rastering by FTIR (Fig. A1).

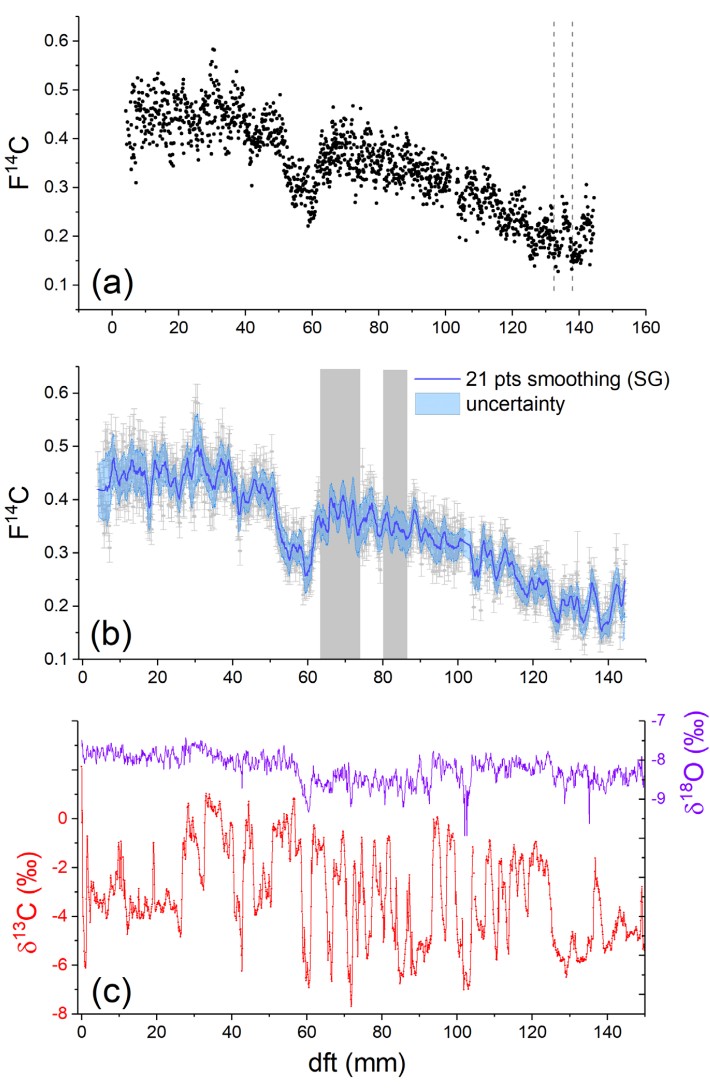

Fig. 2 (A) F$^{14}$C profile of SPA 127 plotted without error bars for reasons of clarity. Grey dashed lines indicate two locations potentially contaminated with epoxy resin (details in SI, Fig.s 3 and 4). (B) F$^{14}$C profile including error bars corresponding to measurement precision (grey) with an overlain SG filter of 21 points interval width with a maximum polynomial degree of 2. The grey shaded areas represent depths in which the quality check for the SG filter was not satisfactory (compare Fig. A1). Details can be found in the text. (C) δ$^{13}$C and δ$^{18}$O signal of SPA 127.

## Results

1. Radiocarbon

$^{14}$C results for both pieces of SPA-127 (T1, T2 and B3) are shown in Fig. 2A, where error bars are not displayed for reasons of clarity. The measured F$^{14}$C ranges from about 0.55 at the top to approximately 0.15 towards the oldest part of the stalagmite. Between 0 and ca. 40 mm, the F$^{14}$C varies around a comparably constant value of 0.45, followed by a pronounced dip between 40 mm and 70 mm. From





70 mm to 140 mm, the F$^{14}$C decreases from 0.4 to 0.2. The measurement precision is approximately
8% for the younger slab and 10% for the older piece with a spatial resolution of 280 µm per data point.
In Fig. 2B, F$^{14}$C data is depicted in grey with uncertainties corresponding to the analytical error. In order
to reduce noise a SG filter has been applied with an interval length of 21 points and the polynomial
order of 2.

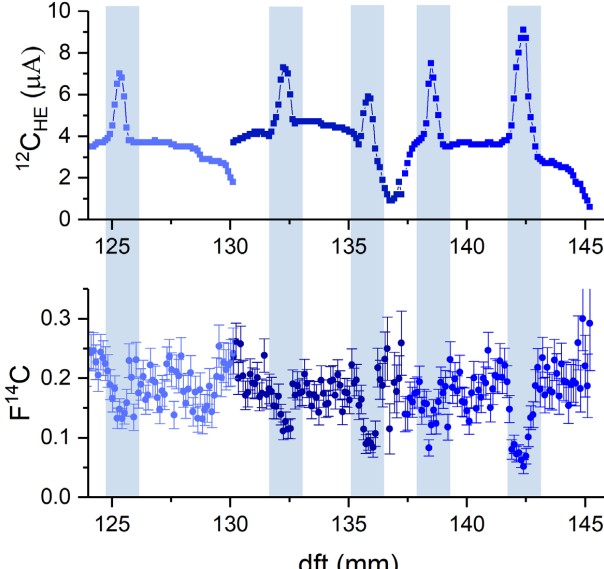


*Fig. 3 Anomalies observed in the signal intensity ($^{12}C_{HE}$) and F$^{14}$C for the initial scan performed on the bottom piece. The scan*
*consists of three sub-scans represented by the different shades of blue. At the beginning and towards the end of each sub-*
*scan the $^{12}C_{HE}$ current rises and drops respectively, i.e. at 130 mm, 137 mm and 145 mm, which is an expected behavior. The*
*anomalies in both measurement parameters are indicated by the blue boxes.*
An unexpected signal was observed in the bottom piece as shown in Fig. 3. The signal intensity (top
panel) recorded for the bottom section of SPA 127 showed five peaks with corresponding F$^{14}$C dips.
The repeated scan (B2), which was performed after removal of the top surface layer showed a similar
behavior (see Fig. A3). We additionally investigated the regions showing these peaks with FTIR. These
anomalies are less distinctly observed in the third LA-AMS scan (see Fig.s A2 and Fig. A4). A detailed
description can be found in the SI.

2. Dead carbon fraction
From the $^{14}$C profile (Fig. 2 B), the Stal-Age (Scholz and Hoffmann, 2011) age-depth model applied on
previously published U-Th data (Fohlmeister et al., 2013) and the known $^{14}$C content of the atmosphere
during the Holocene (Reimer et al., 2013), the dcf was calculated according to equation (1) for the
more than 1500 radiocarbon data points (Fig. 4 A). Again, a SG filter was applied (interval: 21,





maximum polynomial order: 2) and for comparison the $\delta^{18}O$ and $\delta^{13}C$ data are shown in the same graph
(Fig. 4 B). The U-Th-dates and the corresponding average growth rate calculated in Stal-Age (Scholz
and Hoffmann, 2011) is displayed in Fig. 4 C.

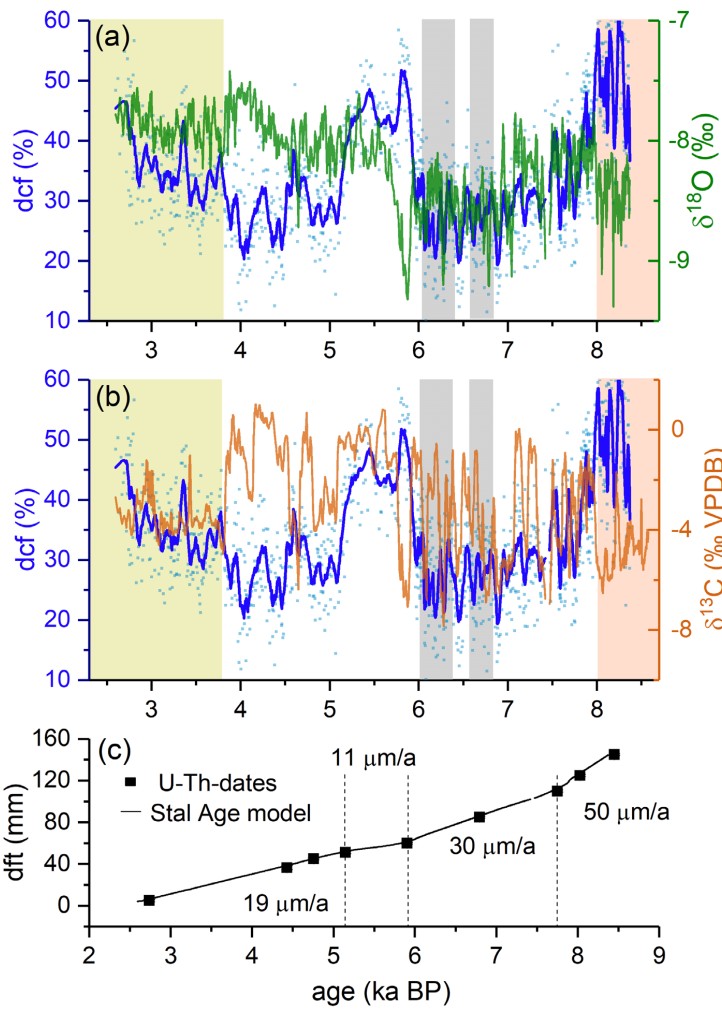


*Fig. 4 Dcf (light blue dots) with a 21 point SG filter (dark blue line) plotted against age and compared to (A) $\delta^{18}O$ (green line)*
*and (B) $\delta^{13}C$ (orange line). The yellow, white and orange shaded areas represent phases with distinct stable isotope and dcf*
*characteristics. Details can be found in the text. Light grey shaded areas mark regions where the SG filter has been determined*
*with lower confidence. (C) Growth history of SPA 127 obtained by StalAge applied on dated depths (black squares). Numbers*
*represent average growth rates in the individual sections.*

3.  FTIR measurements

In the regions where LA-AMS anomalies were observed, two different types of matrices were revealed
by the FTIR spectra (Fig. A2), representing calcite and epoxy resin. The epoxy resin spectra were found
for measurement points along a crack present in this area. The calcite matrix was found in all other
measurement points outside of the crack.
4.  Stable C isotopes
The $\delta^{13}C$ and previously published $\delta^{18}O$ values are shown in Fig. 2C. A large amplitude and fast changes
ranging from -8‰ to +1‰ characterize $\delta^{13}C$ throughout the entire length of the speleothem but are
especially pronounced between 30 and 130 mm. Depths exhibiting a comparably stable $\delta^{13}C$ are found
at the top and bottom of SPA 127, specifically ranging from 0 to 25 mm and from 125 to 150 mm.

## Discussion

The interpretation of the results on C isotopes in SPA 127 will be divided in four main parts: first the
anomalies in $^{14}C$ and $^{12}C$ current observed in the bottom part will be discussed followed by a detailed
discussion on carbon isotope evolution. For this task we subdivided the stalagmite into three sections,
which were identified to underlie different dynamics.
1.  LA-AMS anomalies in the old section of SPA 127 (> 8 ka BP, >120 mm)
The five $^{12}C$-current peaks correlating with strongly depleted $F^{14}C$ observed between 120 and 145 mm
depth, i.e. from 8.0 to 8.4 ka BP indicating that these layers are composed of a different material than
the bulk. The higher $^{12}C$-currents are associated with a matrix that converts more readily into $CO/CO_2$
upon LA compared to $CaCO_3$, a behavior that is known for organic substances with a higher
oxygen/carbon stoichiometric ratio (Frick and Günther, 2012). In order to ascertain whether the
substance in these layers is inherent to the stalagmite or a contamination, its exact composition has
been determined using FTIR. These measurements revealed that the anomalies were caused by epoxy
resin (Fig. A2 and A3).
Indeed, the stalagmite was glued in this section after it broke into two parts shortly after its removal
in 2002 CE. Although the speleothem was only glued on one location, we observed the glue at least at
five positions. Those observations fall in line with small cracks. This suggests, that the glue was able to
soak relatively long distances through those small cracks. Hence, attention should be paid when
working with glued speleothems, especially, if the type of geochemical analyses is based on methods
not easily able to differentiate the analyzed material such as e.g. laser ablation IRMS (Spötl and Mattey,
2006), while e.g. IRMS based on drilled material, which is acidified by $H_3PO_4$, should be unaffected.



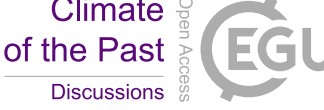

These findings underline that unlike conventional analytical methods applied to stalagmite samples for
$^{14}$C analysis, which provide exclusively the isotope composition of the $CaCO_3$, LA-AMS additionally
yields the $^{14}$C content of OM captured in stalagmites. Despite the fact that this offers novel possibilities
it also requires particular caution to distinguish between inherent OM and contaminants of organic
origin.
$^{14}$C data used for interpretation of SPA 127 stem from scan B3, which was largely unaffected from the
epoxy because the scan was placed off the glued joint as confirmed by conventional $^{14}$C analysis (see
Fig. A4).
2.   Old section of SPA 127 (> 8 ka BP)
In the oldest part of SPA 127, the dcf is comparably high (~60%), while $\delta^{13}$C is relatively depleted with
values smaller than -5‰ on average. Although the reservoir effect is extremely high, in principle the
obtained C-isotope composition can be explained without a major contribution of pyrite oxidation. The
relatively low $\delta^{13}$C value actually contradicts this mode of host rock dissolution but is in line with a
sparse C3 vegetation ($\delta^{13}$C ≈ -25‰) above the cave and nearly completely closed carbonate dissolution
conditions of the host rock (compare Fig. 5 C and D). The stoichiometry of $CaCO_3$ dissolution by
carbonic acid predicts that only about half of the C in the solution comes from the host rock under
nearly completely closed conditions. Under the reasonable assumption that the $F^{14}$C of the host-rock
is 0, the biogenic component must be older than the contemporaneous atmosphere to allow dcf values
larger than 50%. Thus, in addition to living vegetation, which contributes atmospheric radiocarbon, an
old OM source, which respires radiocarbon-depleted $CO_2$, is required to explain depleted $\delta^{13}$C values
and elevated dcf. Such old OM is also argued to have contributed to the radiocarbon reservoir effect
of Moomi Cave (Socotra Island) during the last glacial period (Therre et al., 2020). Observations from
other cave and karst systems also point to important C pools deep in the vadose karst, e.g., (Benavente
et al., 2010; Bergel et al., 2017; Breecker et al., 2012). Nevertheless, in a high elevation and sparsely
vegetated region, this is the first finding of that kind.
Warmer periods favor microbial organic decomposition of, e.g., OM present in the karst below the soil
zone, which leads to an increase in $pCO_2$ and, hence, more acidic water. In turn, more $CaCO_3$ can be
dissolved resulting in higher growth rates, which is observed in this growth period of the speleothem
(Fig. 4). Indeed this phase falls in the period of the early Holocene thermal maximum, which is also
reflected by the depleted $\delta^{18}$O values hinting towards warmer temperatures (Mangini et al., 2005,
Fohlmeister et al., 2013).





CARBONATE DISSOLUTION MECHANISM ALTERNATING BETWEEN SOIL CO₂ AND SULFIDE OXIDATION

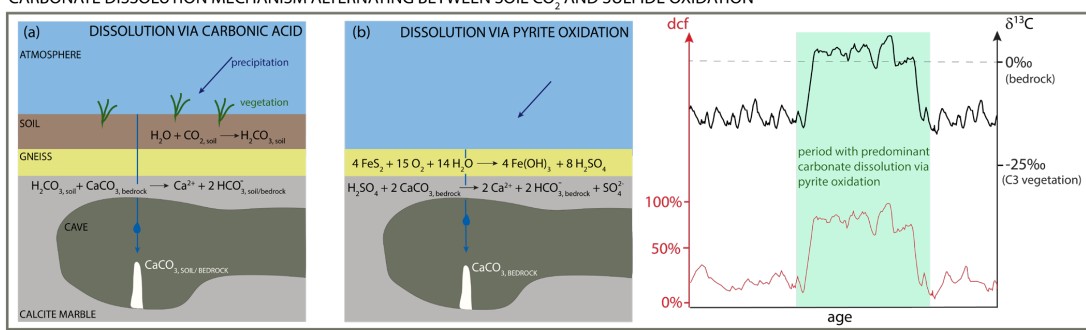

CARBONATE DISSOLUTION MECHANISM: OPEN vs. CLOSED

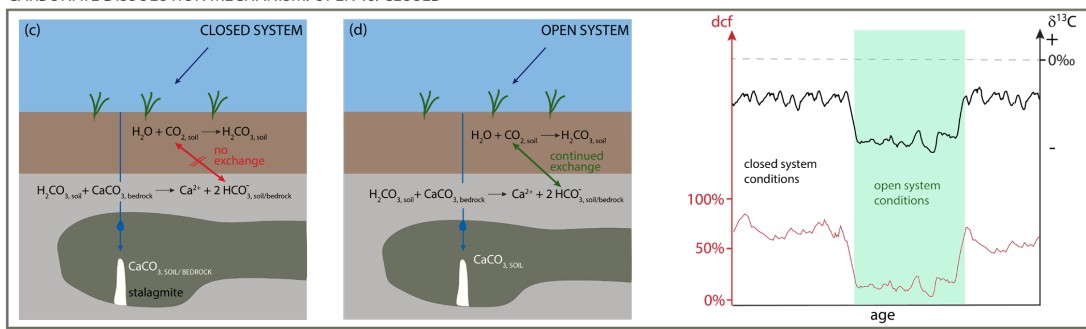

PROCESSES IN CAVE: FRACTIONATION

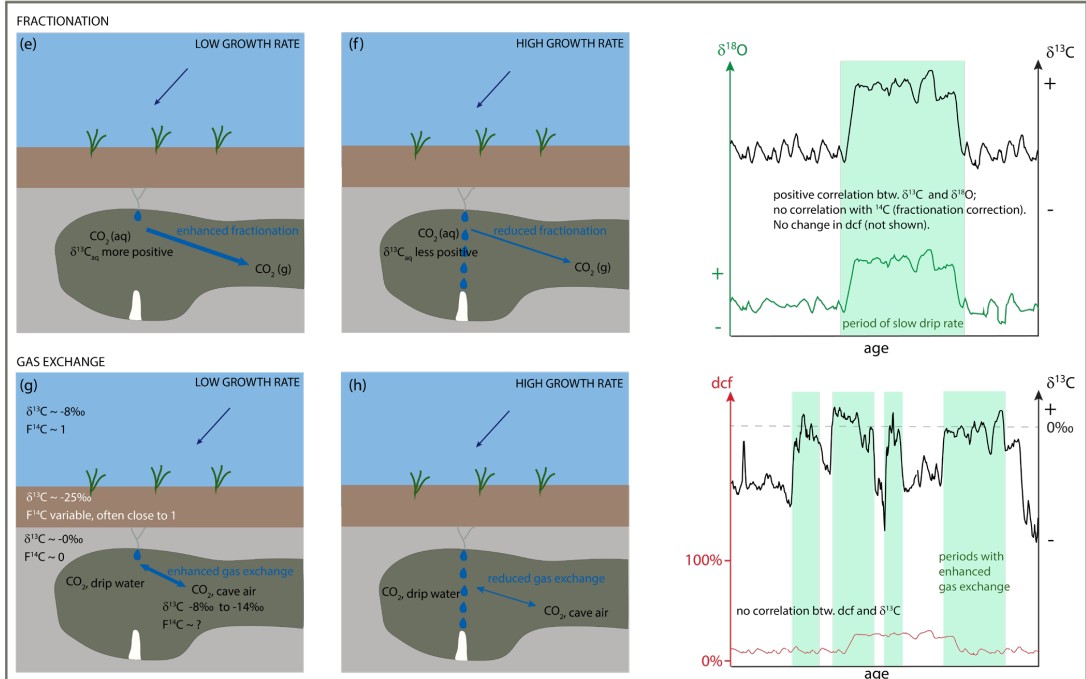


*Fig. 5 Overview of different processes that can influence F¹⁴C and δ¹³C in speleothems. Details can be found in the text.*



3.    Rapid changes in dcf and stable C isotopes (3.8 to 8 ka BP)
As indicated by the reduced growth rate in SPA 127 (Fig. 4), the climate shifted towards a regime with
reduced recipitation. The low $\delta^{13}$C-values of the first growth period are superseded by rapid and very
large variations. This $\delta^{13}$C-pattern is complex and its interpretation is difficult, as this behavior has not
been observed elsewhere. Processes in the soil and karst above the cave as well as in-cave processes
have to be taken into consideration. High-resolution LA-AMS $^{14}$C measurements in conjunction with O
isotope data and growth rate changes, however, greatly assist in disentangling the driving
mechanism(s) for the variations in the C time series. The dcf between 3.8 and 8 ka BP is generally lower
than in the older section, as the additional source for carbonic acid from the old OM in the karst is
most likely strongly reduced either due to the possible reduction in precipitation (as deduced from
growth rate reduction) or that the aged OM reservoir was depleted. The only C sources in this period
are close to modern SOM and the radiocarbon-free host rock.
**Hypothesis 1:** processes above the cave, i.e. different carbonate dissolution processes, cause the rapid
switching
Two different processes may have caused carbonate dissolution at Spannagel cave in this period, i.e.
(i) via carbonic acid formed from soil $CO_2$ and (ii) due to sulfuric acid from pyrite oxidation (compare
Fig. 5 A and B). During times when the first process dominates, the stable carbon isotopic composition
in the stalagmite is strongly influenced by C from the soil shifting the $\delta^{13}$C towards more negative
values. At the same time, the dcf is expected to be relatively low (at least <50%) as the comparably $^{14}$C-
rich soil C contributes to the signal. In contrast, pyrite oxidation leads to more positive $\delta^{13}$C in the
stalagmite corresponding to the $\delta^{13}$C composition of the host rock, as the $\delta^{13}$C depleted biogenic
source contributes little or even no C. Under these conditions, the dcf should increase to values close
to 100% if the modern soil contribution is absent. If the observed $\delta^{13}$C variations are caused by rapid
alternation between both processes, a positive correlation between $\delta^{13}$C and the dcf has to be
expected, with extreme values in the dcf as outlined above. However, no long-lasting, significant
positive correlation for the two data series, i.e. $\delta^{13}$C and dcf, is observed (compare Fig A7) and the
extreme DCF values are not detected. A robust comparison of both data sets is impeded because of
the different spatial offset of the two measurement tracks from the growth axis. The radiocarbon track
is located ca. 5 mm further from the central growth axis than the stable isotope track (Fig. 1). Growth
layers cannot be seen, but the small stalagmite diameter suggests steeply sloping layers resulting in an
apparent shift of the dcf towards older ages relative to the $\delta^{13}$C. Indeed, such a delay is observable
when comparing details in the two data series (Fig. 4 b). Since the curvature of the growth layers is
most likely variable, a constant correction factor cannot be applied. Taking these alignment difficulties
into account, a positive correlation between main features the $\delta^{13}$C and dcf are observable for the



middle period, especially between 3.8 - 5 ka and 6 -8 ka BP. The large changes in $\delta^{13}$C suggest a change
between the carbonate dissolution mechanisms from carbonic acid dissolution to pyrite oxidation.
However, this is expected to be accompanied by an increase of the dcf to 100%, which is not observed.
Generally, the dcf is even smaller than in the youngest and oldest section of the stalagmite, i.e. before
3.8 ka and after 8 ka BP. The correlation between dcf and $\delta^{13}$C suggests that there might have been a
change between the open to closed carbonate dissolution conditions, but the magnitude of $\delta^{13}$C
variations found in SPA 127 is larger than if triggered by this process even when changing from a
completely open to a completely closed system (Hendy, 1971; Fohlmeister et al., 2011). Thus,
additional processes in the cave most likely caused this unusual behavior and the high-magnitude $\delta^{13}$C
variations.
• **Hypothesis 2:** processes in the cave cause the rapid switching
A major factor influencing the carbon isotope composition of stalagmites are fractionation processes
(compare Fig. 5 E and F), which occur during degassing of $CO_2$ and precipitation of $CaCO_3$ from the
solution. Fast dripping results in fast growth and leaves less time for fractionation processes or C
exchange (compare Fig. 5 G and H) and vice versa (e.g. (Fohlmeister et al., 2018; Scholz et al., 2009). In
addition, the difference in $pCO_2$ between water and cave-air $CO_2$ can also influence isotope
fractionation and C exchange processes. In the middle part of SPA 127, the average growth rate
decreased to ≤ 30 μm/a, which is significantly lower than in the oldest section. As discussed earlier,
this might have been partly induced by reduced precipitation resulting in slower drip rates and reduced
growth. In addition, a lower or absent contribution of an old OM reservoir in the karst would have led
to a lower $pCO_2$ difference between the $CO_2$ concentration in the drip water and cave air, which favors
an increase in $\delta^{13}$C through fractionation and C exchange processes in the cave. We hypothesize that
the rapid changes in $\delta^{13}$C might correlate with short-scale changes in growth rate, which cannot be
resolved by the available U-Th chronology, enabling or disabling processes in the cave, i.e. (i)
fractionation and (ii) gas exchange that are described in the following two paragraphs.

(i)      Fractionation effects

During periods of slow growth, fractionation processes can significantly alter the isotopic

composition of the stalagmite. During $CO_2$ degassing from the drip water, the lighter molecules

change preferentially into the gas phase, leaving solution behind that is enriched in heavy

isotopes. This is valid for $^{13}$C and $^{14}$C isotopes. Indeed, recent experiments (Fahrni et al., 2017)

support earlier findings that fractionation of radiocarbon relative to $^{12}$C is about twice as large as

for $^{13}$C relative to $^{12}$C (Stuiver and Robinson, 1974). However, as radiocarbon measurements are



corrected for fractionation effects via $\delta^{13}C$ values, it is impossible to detect a potential correlation
between the two isotopes due to fractionation effects. However, potential fractionation effects
affecting $\delta^{13}C$ also affect $\delta^{18}O$ and can be verified by a positive correlation between stable C
isotopes and O isotopes, e.g. (Dreybrodt, 2008; Polag et al., 2010). Applying a running correlation
coefficient between $\delta^{13}C$ and $\delta^{18}O$ is a powerful tool to detect fractionation changes through time
(Fohlmeister et al., 2017). The 11-point running correlation coefficients calculated for the two
time series of SPA 127 show no stalagmite sections with a high correlation coefficient, but vary
without any obvious pattern between -1 and +1 (compare Fig. A7, bottom panel). Thus,
fractionation was most likely not the main process causing the large variations in $\delta^{13}C$, but may
have played a minor role during some periods.
(ii)      Gas exchange processes
Another process that may be dominant if the stalagmite growth rate is sufficiently low is C
exchange between $CO_2$ of the cave air and C dissolved in the drip water. This becomes important
if drip intervals are long and/or the differences between the $pCO_2$ of the water and cave air is
small (Hendy, 1971; Scholz et al., 2009). In this case, the C isotopic composition of the drip water
when reaching the top of the stalagmite depends mainly on the initial $\delta^{13}C$ of drip water and on
the degree of C exchange with the cave atmosphere. Spannagel Cave is well ventilated throughout
the year with cave air $\delta^{13}C$ values of -10 to -11‰ (Töchterle et al., 2017), which is significantly
lighter than that of the atmosphere, i.e. approximately -8‰. These more negative values are a
hint towards a contribution from soil air to the cave air, reaching the cave through cracks in the
bedrock. The following assumptions were made: the $\delta^{13}C$ of drip water is composed of two
biogenic C sources ($\delta^{13}C \sim -25‰$) and host rock ($\delta^{13}C \sim 0‰$) and about 20 - 30% are derived from
the host rock (based on the dcf in this interval). Accounting for about 10‰ to 11‰ fractionation
between soil gas $CO_2$ and $HCO_3^-$ during the transition of soil gas $CO_2$ to dissolved inorganic carbon
(DIC), the initial drip water, which was feeding the stalagmite, has a $\delta^{13}C$ value between -12 and -
11‰. Considering progressing Rayleigh fractionation effects in the cave, carbonate $\delta^{13}C$ values of
-8‰ appear feasible (Scholz et al., 2009, Deininger et al., 2012) without any exchange of C. C
exchange processes lead to water significantly enriched in $^{13}C$. When cave air of $\delta^{13}C$ around -11‰
exchanges with drip water, the C isotopic composition of the water will increase, as the transition
of gaseous $CO_2$ to $HCO_3^-$ involves a fractionation of about +10 to +11‰ at temperatures between
0 and 5°C. Thus, drip water in C isotopic equilibrium with cave air $CO_2$, which is the most extreme
case, should have $\delta^{13}C$ values of -1 to 0‰. Precipitation of $CaCO_3$ from such water would result in
$\delta^{13}C$ values of around 0 to +1‰ as observed for some short periods.





Similar assumptions regarding the cave air with respect to $\delta^{13}C$ can be applied to radiocarbon. If
cave ventilation is sluggish, $F^{14}C$ in the cave air will deviate from atmospheric values (i.e. $F^{14}C_{atm}\approx1$)
as it is influenced by other sources, such as transferred soil air or degassed $CO_2$ from drip water,
which both are depleted with respect to atmospheric values. In a recent study, cave air has been
shown to be depleted in radiocarbon with values as low as $F^{14}C\approx0.6$ (Minami et al., 2015). When
cave ventilation is not effectively changing the depleted radiocarbon towards more atmospheric
values, C isotope exchange processes are not detectable from $^{14}C$ in speleothems. As isotopic
fractionation is not important for radiocarbon (as explained above), C isotope exchange of DIC
with cave air, which both have a similar radiocarbon content, would have no effect on the $^{14}C$
signal of the precipitated calcite.
In summary, combined high-resolution $\delta^{13}C$ and radiocarbon measurements are a valuable tool to shed
new light on processes affecting C isotopes in the subsurface. Using well justified assumptions and
first-order calculations of mixing and fractionation effects, in-cave C-exchange processes remain the
only explanation for the rapid $\delta^{13}C$ anomalies.

4.    Interpretation of dcf and $\delta^{13}C$ in the youngest section of SPA 127 (2.5 to 3.8 ka BP)
In the youngest section of this stalagmite a behavior similar to the oldest section with respect to $\delta^{13}C$
and radiocarbon content is observed. From approximately 3.8 ka BP on, the dcf increases slowly from
about 20% to 50%. Correspondingly, $\delta^{13}C$ shows a lower variability than in the middle part with mean
$\delta^{13}C$ values of -3‰ to -4‰, which is also comparable to the behavior observed for the interval
> 8 ka BP. As $\delta^{13}C$ is not showing any long-term trend as observed for the reservoir effect, we rule out
a change to more closed carbonate dissolution conditions driving the increase in dcf. The only
explanation that can lead to such an increase is the development of an "old" C reservoir. We propose
that climatic conditions have changed such that this old OM pool in the karst, which is decoupled from
the atmosphere, successively contributed $CO_2$ to the stalagmite $CaCO_3$ until it reached similar values
as in the period before 8 ka BP. Between approximately 8 and 6 ka BP the Alps experienced a warmer
climate than today (Ivy-Ochs et al., 2009; Nicolussi et al., 2005). A study conducted by Nicolussi et al.
(2005) in the Kauner valley, situated approximately 70 km west of Spannagel Cave, showed that the
timberline was significantly higher during that period (Fig. 6) supporting a warmer climate. Between 6
and 4 ka BP the timberline was comparable with the present day situation.
It is expected that with the lowering of the timberline after ~4 ka the vegetation density decreased as
well, which should be reflected in the cave carbonate $\delta^{13}C$. This, however, is not the case during this
period pointing towards a relatively stable contribution of organic derived $CO_2$. Possibly, a certain

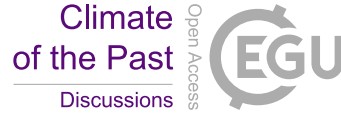

proportion of plants that grew during the early to mid-Holocene warm epoch died and the
corresponding OM located in the deeper vadose zone was initially stabilized due to reduced
precipitation and later became mobilized due to enhanced microbial activity. Considering the low
mean annual temperatures at the high-alpine site, decomposition processes are most likely slow,
allowing OM to ageing during decomposition as indicated in a recent study by Shi et al. (2020) . The
radiocarbon composition of the ageing SOM will closely follow a radiocarbon specific exponential
decay and is responsible for a depleting soil gas $CO_2$. Depending on the contribution of root-respired
$CO_2$ of living plants (grasses, alpine mats) compared to the decomposed $CO_2$ of dead OM, the initial
$F^{14}C$ will closely follow an exponential decay, resembling that of the radiocarbon decay and thus would
contribute to the observed increase of the dcf. The closer the observed decrease in the initial $F^{14}C$
follows that of radiocarbon the larger the contribution of $CO_2$ from the aging SOM reservoir. Based on
the observed rate of decrease in initial $F^{14}C$, which compares well with the radiocarbon decay trend
(Fig. 6), we suggest that the majority of $CO_2$ that contributed to the speleothem $CaCO_3$ must have had
its origin in ageing soil OM.

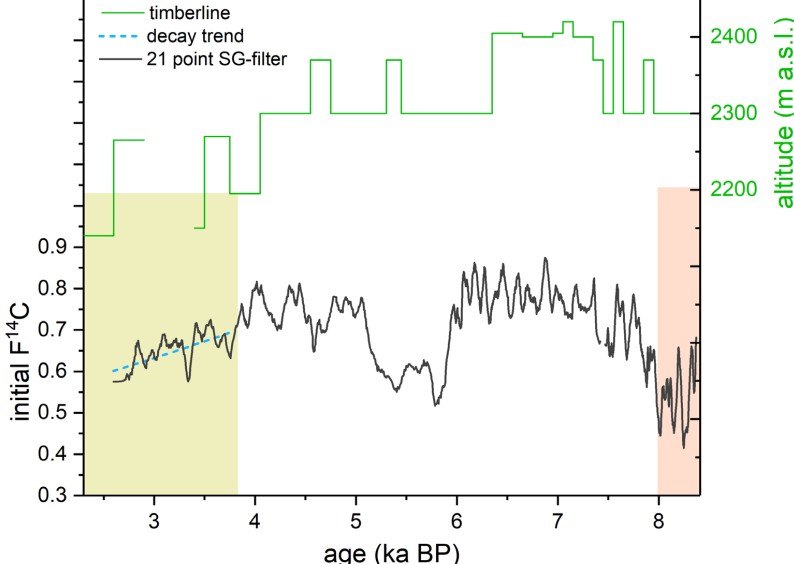


*Fig. 6 Comparison of the timberline reconstruction of the nearby Kauner valley (green line, after Nicolussi et al. (2005)) with*
*the initial $F^{14}C$, i.e. the $^{14}C$ present in the respective layer at the time when it formed SPA 127. Around 4 ka BP the timberline*
*starts to decline which is concurrent with a decrease in initial $F^{14}C$. This decrease closely follows a radiocarbon decay trend*
*(blue line). Green and red areas mark the three different time periods as indicated in Fig. 4.*



## Conclusion

Combined stable carbon isotope and radiocarbon analysis of stalagmite SPA 127 provide a comprehensive picture on the carbon dynamics at the high-alpine Spannagel Cave. Due to the novel LA-AMS technique, a high-spatially resolved $^{14}$C time series allowed unprecedented insights into processes in this karst system. Care has to be taken when applying LA-AMS to stalagmites as epoxy resin used in sample preparation leads to distorted results.

Key findings of this study for the three periods characterized by different C dynamics:

(i) > 8 ka BP: generally low and stable $\delta^{13}$C values combined with a comparably high dcf (>50%) point towards the existence of an old OM reservoir in the karst rock, which provides additional carbonic acid and enhances bedrock dissolution.

(ii) 3.8 – 8 ka BP: this period is characterized by a strong variability in $\delta^{13}$C with a generally lower dcf suggesting that the old OM reservoir in the karst had either been exhausted or stabilized (less production to aged respired soil/karst $CO_2$) possibly due to reduced precipitation. This is supported by a lower stalagmite growth rate in this period. In-cave *gas exchange processes* are the most likely explanation for the strong $\delta^{13}$C variability, as (i) *bedrock dissolution* mechanisms, i.e. pyrite oxidation vs. carbonic acid dissolution, are not supported by the magnitude of changes in dcf and stable C, even though the temporal coherence indicates that some of the $\delta^{13}$C variations might be explained by the bedrock dissolution mode (open vs closed carbonate dissolution system) and, (ii) *fractionation processes* in the cave cannot explain the large shifts as no correlation between $\delta^{18}$O and $\delta^{13}$C is observed.

(iii) 2.4 – 3.8 ka BP: the comparably more stable $\delta^{13}$C signature combined with an increasing dcf hints towards contribution of an ageing OM reservoir in the karst similar to the period > 8 ka BP. This OM reservoir contributed to the stalagmite growth in this period due to warmer climatic conditions. While the contribution of old OM in the oldest growth phase was stable, the youngest section indicates an ageing of this OM reservoir.

## Author contribution

CW, JF and CS conceptualized the content of this manuscript. CW, MW, BH carried LA-AMS measurements out, CS conducted stable carbon isotope analyses. MW and LW developed the data reduction strategy. JF, CW and TE interpreted the data and compared them to published records. CW prepared the manuscript with contributions from all co-authors.



## Competing interests

The authors declare that they have no conflict of interest.

## Acknowledgements

JF acknowledge support from DFG grants FO 809/2-1 and FO 809/4-1. MW was supported by ETH
Research Grant ETH-03 18-2. We thank Laura Hendriks for performing the FTIR analyses.



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

cave drip waters in Ryugashi Cave, Shizuoka Prefecture, central Japan." Nuclear Instruments and
Methods in Physics Research Section B: Beam Interactions with Materials and Atoms 362: 202-209.
https://doi.org/10.1016/j.nimb.2015.05.020
Nicolussi, K., Kaufmann, M., Patzelt, G., Plicht van der, J. and Thurner, A. (2005). "Holocene tree-line
variability in the Kauner Valley, Central Eastern Alps, indicated by dendrochronological analysis of living
trees and subfossil logs." Vegetation History and Archaeobotany 14(3): 221-234. 10.1007/s00334-005-
0013-y
Noronha, A. L., Johnson, K. R., Hu, C., Ruan, J., Southon, J. R. and Ferguson, J. E. (2014). "Assessing
influences on speleothem dead carbon variability over the Holocene: Implications for speleothem-
based radiocarbon calibration." Earth and Planetary Science Letters 394: 20-29.
https://doi.org/10.1016/j.epsl.2014.03.015
Noronha, A. L., Johnson, K. R., Southon, J. R., Hu, C. Y., Ruan, J. Y. and McCabe-Glynn, S. (2015).
"Radiocarbon evidence for decomposition of aged organic matter in the vadose zone as the main
source of speleothem carbon." Quaternary Science Reviews 127: 37-47.
10.1016/j.quascirev.2015.05.021
Polag, D., Scholz, D., Mühlinghaus, C., Spötl, C., Schröder-Ritzrau, A., Segl, M. and Mangini, A. (2010).
"Stable isotope fractionation in speleothems: Laboratory experiments." Chemical Geology 279(1-2):
599   31-39
Reimer, P. J., Bard, E., Bayliss, A., Beck, J. W., Blackwell, P. G., Ramsey, C. B., Buck, C. E., Cheng, H.,
Edwards, R. L., Friedrich, M., Grootes, P. M., Guilderson, T. P., Haflidason, H., Hajdas, I., Hatté, C.,
Heaton, T. J., Hoffmann, D. L., Hogg, A. G., Hughen, K. A., Kaiser, K. F., Kromer, B., Manning, S. W., Niu,
M., Reimer, R. W., Richards, D. A., Scott, E. M., Southon, J. R., Staff, R. A., Turney, C. S. M. and van der
Plicht, J. (2013). "IntCal13 and Marine13 Radiocarbon Age Calibration Curves 0–50,000 Years cal BP."
Radiocarbon 55(4): 1869-1887. 10.2458/azu_js_rc.55.16947
Richards, D. A. and Dorale, J. A. (2003). "Uranium-series chronology and environmental applications of
speleothems." Rev. Mineral 52: 407-460. 10.2113/0520407
Savitzky, A. and Golay, M. J. E. (1964). "Smoothing and Differentiation of Data by Simplified Least
Squares Procedures." Analytical Chemistry 36(8): 1627-1639. 10.1021/ac60214a047



Scholz, D. and Hoffmann, D. (2008). "²³⁰Th/U-dating of fossil reef corals and speleothems." Quater. Sci.
J. 57(1-2): 52-77
Scholz, D. and Hoffmann, D. L. (2011). "StalAge - An algorithm designed for construction of speleothem
age models." Quat. Geochronol. 6(3-4): 369-382. 10.1016/j.quageo.2011.02.002
Scholz, D., Mühlinghaus, C. and Mangini, A. (2009). "Modelling δ13C and δ18O in the solution layer on
stalagmite surfaces." Geochimica et Cosmochimica Acta 73(9): 2592-2602
Shi, Z., Allison, S. D., He, Y., Levine, P. A., Hoyt, A. M., Beem-Miller, J., Zhu, Q., Wieder, W. R., Trumbore,
S. and Randerson, J. T. (2020). "The age distribution of global soil carbon inferred from radiocarbon
measurements." Nat. Geosci. 13(8): 555-559
Southon, J., Noronha, A. L., Cheng, H., Edwards, R. L. and Wang, Y. (2012). "A high-resolution record of
atmospheric 14C based on Hulu Cave speleothem H82." Quaternary Science Reviews 33: 32-41
Spötl, C. (2011). "Long-term performance of the Gasbench isotope ratio mass spectrometry system for
the stable isotope analysis of carbonate microsamples." Rapid Communications in Mass Spectrometry
623  25(11): 1683-1685
Spötl, C., Fairchild, I. J. and Tooth, A. F. (2005). "Cave air control on dripwater geochemistry, Obir Caves
(Austria): Implications for speleothem deposition in dynamically ventilated caves." Geochimica et
Cosmochimica Acta 69(10): 2451-2468
Spötl, C., Fohlmeister, J., Cheng, H. and Boch, R. (2016). "Modern aragonite formation at near-freezing
conditions in an alpine cave, Carnic Alps, Austria." Chemical geology 435: 60-70
Spötl, C. and Mangini, A. (2007). "Speleothems and paleoglaciers." Earth and Planetary Science Letters
630  254(3-4): 323-331
Spötl, C. and Mangini, A. (2010). "PALEOHYDROLOGY OF A HIGH-ELEVATION, GLACIER-INFLUENCED
KARST SYSTEM IN THE CENTRAL ALPS (AUSTRIA)." Austrian Journal of Earth Sciences 103(2)
Spötl, C. and Mattey, D. (2006). "Stable isotope microsampling of speleothems for
palaeoenvironmental studies: A comparison of microdrill, micromill and laser ablation techniques."
Chem.Geol. 235(1-2): 48-58. 10.1016/j.chemgeo.2006.06.003
Stuiver, M. and Robinson, S. W. (1974). "University of Washington GEOSECS north Atlantic carbon-14
results." Earth and Planetary Science Letters 23(1): 87-90
Therre, S., Fohlmeister, J., Fleitmann, D., Matter, A., Burns, S. J., Arps, J., Schröder-Ritzrau, A., Friedrich,
R. and Frank, N. (2020). "Climate-induced speleothem radiocarbon variability on Socotra Island from
the Last Glacial Maximum to the Younger Dryas." Climate of the Past 16(1): 409-421
Töchterle, P., Dublyansky, Y., Stöbener, N., Mandić, M. and Spötl, C. (2017). "High-resolution isotopic
monitoring of cave air CO2." Rapid Communications in Mass Spectrometry 31(11): 895-900.
10.1002/rcm.7859
Welte, C., Wacker, L., Hattendorf, B., Christl, M., Fohlmeister, J., Breitenbach, S. F., Robinson, L. F.,
Andrews, A. H., Freiwald, A. and Farmer, J. R. (2016a). "Laser Ablation–Accelerator Mass Spectrometry:
An Approach for Rapid Radiocarbon Analyses of Carbonate Archives at High Spatial Resolution."
Analytical chemistry 88(17): 8570-8576
Welte, C., Wacker, L., Hattendorf, B., Christl, M., Koch, J., Synal, H.-A. and Günther, D. (2016b). "Novel
Laser Ablation Sampling Device for the Rapid Radiocarbon Analysis of Carbonate Samples by
Accelerator Mass Spectrometry." Radiocarbon 58(02): 419-435. doi:10.1017/RDC.2016.6
Welte, C., Wacker, L., Hattendorf, B., Christl, M., Koch, J., Yeman, C., Breitenbach, S. F. M., Synal, H. A.
and Gunther, D. (2017). "Optimizing the analyte introduction for C-14 laser ablation-AMS." Journal of
Analytical Atomic Spectrometry 32(9): 1813-1819. 10.1039/c7ja00118e
