# Peer review of "Climatic variations during the Holocene inferred from"

_Climate of the Past, 2020_

## Referee Comment (RC1) · Anonymous Referee #1 · 18 Nov 2020

Summary: Carbon isotopes of speleothems are difficult to interpret because of the several mechanisms that can affect them (changes of plants above the cave, carbonate dissolution mechanisms, in-cave fractionation processes). This study uses a recently developed method of laser ablation coupled to accelerator mass spectrometer to measure high-resolution 14C in a speleothem from western Austria and compares it with stable carbon and oxygen isotopes to explore key processes influencing carbon isotope compositions.

Key findings that I took away from this study: The coupled methodology (novel 14C measurements w/ stable isotopes) reveals changes of the presence of older OM reser-

voirs in the karst rock above the cave through time that are attributing to changing carbon isotope signals:

1. >8ka: there is an older reservoir of OM that causes low $\delta$13C and high dcf. *This interpretation wouldn't have been possible without the 14C measurements because low values of $\delta$13C alone lead to the interpretation of C3 vegetation above the cave...and the measured high dcf is needed to distinguish this from the actual mechanism of vegetation + an older OM reservoirs source that is causing elevated CO2 values.

2. 3.8-8ka: There is strong $\delta$13C variability and lower dcf. This suggests there is not an older source of OM contributing to the carbon isotope signal, and the interpretation is made it was stabilized/exhausted because of reduced precipitation. While this interpretation makes sense, I do not think bringing in growth rate is an accurate argument because I am not sure how "significant" a drop from 19$\mu$m to 30$\mu$m in growth rate is, and a change in growth rate could be from a variety of factors independent of precipitation amount (i.e. chemical kinetic processes, dissolved Ca2+ concentration, etc.). I recommend strengthening your argument on this (are their regional proxy records that suggest there was reduced precipitation amount at this time?) or removing the growth rate stance as a whole. Also, the interpretation is also made that "in-cave gas exchange processes are the most likely explanation for the strong $\delta$13C variability", and bedrock dissolution/fractionation processes are ruled out. There should be clarification for what you mean by "in-cave" gas exchange processes, because right now you make this interpretation, yet right before you rule out bedrock dissolution and fractionation mechanisms. *Also, why are the distinct isotopic changes at ~5 and 6ka not discussed? At both of these times, it appears $\delta$18O and dcf (~6ka), and $\delta$13C and dcf (~5ka) change drastically. Also at 6ka the dcf increases greatly – what could cause this? Why is this not mentioned? It definitely should be if the steady increase in dcf in the younger part of the sample is discussed (I find the sharp changes at 6-5 ka to be more significant than the younger part of the sample).

3. 2.4-3.8 ka: The more stable $\delta$13C signature and increase in dcf suggests a contribution of aging OM reservoir. This old OM reservoir source is suggested to have been from the buildup of a mid- Holocene warm epoch. Overall, I follow this interpretation and thinks it makes sense. How/why is there the rapid decay trend of F14C at ∼6ka?

Comments: Overall, I think this is a nice study and will make a worthwhile contribution to the literature. I think a future of combining novel high-resolution 14C measurements with other stable isotope data will definitely aid in the interpretation of these systems in terms of climactic processes. I think a bit of reworking with the discussion to help overall flow and clarifying several sections will result in a nice manuscript.

Some care should be taken when lumping speleothem "growth rate" with these interpretations – however, as a reduction in growth rate is not always a direct relation to reduced precipitation [i.e. changes in chemical kinetics, flow conditions (turbulent vs. laminar) could also control the growth rate of speleothems]. More of this is noted below, but this was one main issue I had with the interpretations.

I think this paper would strengthen if there was a brief, added section on what the initial 14C (F14C) raw values can tell us, vs. what the dcf tells us. The paper mentions what factors can contribute to dcf (lines 78-92), but it would be good to mention what the initial 14C values can tell us in this section as well. Especially because the F14C is referred to at the end of the paper (during the discussion of the youngest part of the sample, and in Fig. 6), which becomes slightly confusing because the entire paper is centered around the dcf values. I understand the distinction between the two (the use of initial 14C values to calculate for dcf) but adding a sentence to explain why you would look at raw, initial F14C values vs. dcf would be beneficial for overall flow and clarity of the manuscript and interpretations.

In addition, there are lots of data in this manuscript and I'm confused what all your study specifically measured (the radiocarbon) vs. what data from other studies you are comparing it with (i.e. $\delta$18O, $\delta$13C? Did you all measure $\delta$13C?). I would make this clearer in the methods somewhere by stating directly: "in this study, we measured

radiocarbon and compared with XYZ from other studies."

Comments on tables & figures: Table 1 – Why is there a "*" after the "expected" in the table. I see no explanation of this. Also, under the "Expected* 13C" column of the "Old OM contribution to seepage water acidification" row, does the "shift to more negative values" mean <0 or <10? If it's closer to <10, I'd label it like that since that's what you have in the above two rows (for consistency).

Figure 1: There is no ruler in this photo. It says the length in the caption, but it would be helpful to have a ruler for reference. This would especially be helpful when looking back at this figure during the discussion when you say the "old section of SPA 127 (>8,ka, >120mm)", because then we'll be able to go and see where in the sample this is.

Figure 2: Are the $\delta$13C and $\delta$18O from other studies? If so, cite the studies in the figure caption after part (C).

Figure 2: Also, it would be helpful to plot the age in this figure (since age is what you refer to in the discussion, not depth). You could add it by an additional x-axis on the top.

Figure 3: You mention that "12CHE" is the signal intensity in the figure caption, but nowhere in the text do you explain this further. Can you add a section somewhere that says this? It becomes confusing during the discussion section (e.g. lines 270-273), because in these sections you refer to it only as "12C", and not 12CHE".

Figure 4: What stands out to me is the jump at 6ka (increase in dcf, decrease in $\delta$18O, and decrease in $\delta$13C), but this isn't mentioned in the text at all? How come?

Figure 4, overall comment: "the yellow, white, and orange shaded areas represent phases with distinct stable isotope and dcf characteristics..." but what about the two sections (1: ~6ka when dcf increases and dO18 and $\delta$13C decreases; 2: ~5ka when $\delta$13C and dcf drops)? These transitions aren't really talked about in the text, and I'm

wondering why you chose not to select these areas as "phases" with distinct characteristics?

Figure 4c: How do you know there are not hiatuses in between the U-Th ages? For example, it appears right before an age of 4 (the yellow/white boundary in panels 4a and 4b), there is a sharp decrease in dcf, increase in $\delta$18O), and increase in $\delta$13C), yet this is the part of your sample that has the longest gap of age control. How do you know there's not a micro-hiatus here that's undetectable? It may be worth mentioning you can't rule this hypothesis out, just to cover your bases and to let readers know you thought of this (rather than just interpreting this as a purely real signal). Also, the U-Th ages need error plotted.

Comments by line: Line 60: Can you give a few examples in this sentence of what you mean by "in-cave processes"? I see you cite Mattey et al., 2016 and Spötl et al., 2005, but it would be easier for readers to follow what you mean by this by stating it clearly in the sentence.

Line 83-85: You mention the conditions in both an "open vs. closed" system can affect the dcf, but you only describe how the dcf typically is in an open system. I suggest adding a sentence describing what dcf would be in a closed system for clarity. Also detailing how there could be a change from an open to closed system may be helpful. I see you outline it in Figure 5, and also in Table 1, so perhaps just simply referencing these two figures/tables will help streamline this discussion.

Line 88-92: "Several studies. . .more closed-system conditions under higher precipitation regimes". . .maybe reword this sentence because I'm not sure what exactly you mean by it. What constitutes "more closed-system" conditions? Once again, perhaps by referencing either Table 1 or Figure 5 this would help.

Line 136: How do you know there aren't hiatuses present in your speleothem? Perhaps briefly state how you approached assessing hiatuses in your sample here. Also what is the error on each age? I don't see this stated anywhere, and it's not in Figure 4C.

The error should be plotted.

Line 162-163: "offset between stable isotope and radiocarbon data of up to several hundred micrometers", how did you go about accounting for this? Perhaps briefly describe what you did to account for this offset so readers are aware of your methodology. *I see in line 350 that you are unable to apply a correction factor. Perhaps stating this earlier (such as at line 162-163) will help the reader better understand your process of approaching this.

Line 193: It should be stated in the first sentence or two what you used this technique for. Example, directly stating: "FTIR was used for....", because right now it is unclear why you used FTIR (it's not until later in the discussion when you explain identifying the contaminated epoyy area, and I think it'd be better to state up front in this section). *Line 276: "it's exact composition has been determined using FTIR". It is not until this sentence that I realize what you are using FTIR for. Perhaps add a sentence to the FTIR section (the section starting at line 193) that states, "we use FTIR to determine specific compositions of areas in our sample to clarify the causes of anomalies."

Line 243: "For the more than 1500 radiocarbon data points", I suggest just inserting the exact number of data analysis points that you have here, instead of saying "more than.."

Line 259: Please add a reference at the end of this sentence to let readers know where the "previously published $\delta$18O values" can be found. Also, did you measure the $\delta$13C values in this study? Or did you pull data form another study? This is unclear and should be clarified.

Line 265: I think this paragraph could be reworded so it's clearer. I'm a bit confused about how the different sections of the discussion are broken up the way they are. A suggestion: section "1. LA-AMS anomalies in the old section of . . ." should be an entirely separate section than the ones below, because it just details how the presence of epoxy caused contamination, and there are no other interpretations of the data associated with it. This entire section be a brief paragraph at the start of the discussion section, and then you could transition into a "part II" of the discussion that's exclusively about the interpretation of the isotope systems across different parts of the sample.

Line 266: A suggestion to clarify this sentence: "..in the bottom part of the sample"

Line 270: Please add the reference to Figure 3 after this sentence: "The five 12C-cururent peaks correlating with....(Figure 3)" so the readers know this is what you're referencing. Also change "indicating" to "indicate".

Lines 300, 328, 335, 337, 339, 396, 397: Change "C" to "C-isotope". I probably missed additional places you refer to just "C", so please change everywhere this occurs.

Comment on section "2. Old section of SPA (>8 ka BP) (Line 293-315): The first paragraph of this section (line 294-309) walks readers through interpretations of high dcf values and low $\delta$13C values from >8ka. A transitional sentence is needed in the beginning part of the second paragraph (lines 310-315), to set up the connection of the two (i.e. the warmer temperatures for the Holocene could have caused the mobilization of the older OM), because right now it feels out of place a bit. Also adding a concluding sentence would be beneficial to wrap up this section of the discussion.

Line 319: Perhaps state at what depth/age you are referring to here in this sentence. For example: "As indicated by the reduced growth rate in SPA 127 (Fig. 4, 3.8 ka).."

Line 320: Misspell of "precipitation" (it says "recipitatoin").

Line 321: Please add this to the sentence for clarity: "The low $\delta$13C -values of the first growth period are superseded by rapid and very large variations of $\delta$13C."

Line 325-328: You state here "the dcf between 3.8 and 8 ka is generally lower than the older section..." but at ~5.8-6ka dcf jumps relatively high and stays high until ~5.2 ka. I think this should be addressed somewhere in your discussion.

Line 343, 392: what is (Fig A7?) Do you mean supplemental Figure 7 (Fig S7)?

Line 330: Hypothesis 1, general comments – Line 351: "a positive correlation between main features of $\delta$13C and dcf are observable for the middle period, especially between 3.8-5 ka and 6-8 ka BP." I would argue @ 4ka they appear reversed, but 6-8ka I believe I see this correlation. A suggestion: zoom in on these two time slices for what's plotted in Figure 4B, especially the 6-8 ka one, so the correlation is clearer (the blue and orange lines are kind of on top of each other in the figure now so it's hard to see...). The argument for this could be stronger if you could demonstrate the relationship more clearly. Line 354: A bit more explanation for why "an increase of the dcf to 100%" is needed for this mechanism to work would help the flow of this argument better. Line 355: "Generally, the dcf is even smaller than in the youngest and oldest section of the stalagmite.." What about at from 6-5ka? This need addressed.

Line 354: "..this is expected to be accompanied by an increase of the dcf to 100%." Why? Some elaboration on this would strengthen your argument.

Line 362: Hypothesis 2, general comments – Overall the text in the discussion of this hypothesis is clear. But I disagree with your growth rate argument. As stated in previous comments, I'm not sure if a change in the growth rate ($19\mu$m/$11\mu$m to $30\mu$m ) is "significant" – I consider this just a "lower" growth rate. I therefore don't think you can use this piece of information to suggest it was caused by an overall reduction in precipitation amount. I think you should either try to bring in literature that demonstrate regional drier conditions, or some other support for this argument other than growth rate.

Lines 395-427: I'm a bit confused with this paragraph. I follow your discussion, but are you saying this is your main interpretation for what is happening during this interval and causing all the fluctuations? (i.e. is this what you mean by "in-cave" processes). You state in the conclusion that it's not bedrock dissolution or fractionation processes, so are you interpreting the strong variability in $\delta$13C as a change of gas exchange processes? If so, this needs to be stated clearer, because right now it's a bit ambiguous whether you mean this or not.

---

## Referee Comment (RC2) · Anonymous Referee #2 · 11 Feb 2021

Review of CP-2020-110 "Climatic variations during the Holocene inferred from radiocarbon and stable carbon isotopes in a high Alpine cave" by Welte et al.

**General comments:**

This paper could be of great interest to scientists challenged with properly disentangling the controlling factors of d13C in caves, which is quite complex as has been reviewed recently by Fohlmeister et al. (2020, GCA).

The strength of the paper is the combination of radiocarbon concentration and d13 C in stalagmites samples at a continuous and high spatial resolution to understand what happens above the Spannagel cave during some period of the Holocene. However, I found that the manuscript is moderately written (please don't take this personal, and as a reviewer I am not qualified to comment in detail about the writing, but I strongly believe there is a room for improvement and writing is a learning process). Also, in my opinion, the manuscript requires a lot of reorganization of ideas to make it clearer. For example, the first section of the discussion section, the anomalies in the old section of SPA 127 do not highlight the importance of the proxies being studied. Instead, it discusses the result based on methodological approach. Please note that the journal "Climate of the Past" is not a methodology journal (if this paper was submitted to a method paper, then I would not argue about having this part in the discussion). In fact, I see that this whole section either belongs to the method or some part of it could go as a quick interpretation of the data in the result section, but it should not belong to the discussion section. Another example, a separate generality section about dcf can be helpful here (see detailed comments) where the authors could explain the difference between radioactive 14C and dead carbon, and what are the potential sources of them. With this said, the sections at L 67-105 could belong to that independent section right between the introduction and the Materials &methods. If that general&fundamental notion is separated from the introduction, I am certain that the introduction could become concise and clearer, with a clear statement of the problematic, and a proposition of the new method and its potential relevance in future paleo– reconstruction.

I also feel the title does not fully capture the content of the manuscript. While the authors entitled their manuscript "Climatic variations during the Holocene inferred from

radiocarbon and stable carbon isotopes in a high Alpine cave", I found that the manuscript mainly use radiocarbon and d 13 C as a proxy for local changes and specifically what happens right above the cave in the epikarst, and not directly to climate. In their conclusion, it was made clear that these two are good proxies to understand carbon dynamic. Hence, I think the authors should emphasize the importance of d13C and F14C in the use of stalagmites in paleoenrivonmental reconstruction and build their discussion based on that, rather than jumping directly to climate, which at this stage seems more speculative.

There are several points by the authors in manuscript that support my comments. For example, in the abstract, the authors used the variation in 14 C and d13C as an evidence of host bedrock dissolution or organic matter reservoir contribution from the epikarst to the cave.   And in fact, this has been one of the focuses of the interpretation/discussion. The authors should make that clear that from using such inferences, information from the local place can later be applied to climatic context. In my reading of the manuscript, the bridge '*local response–climate*' is quite obscure (possibly by the current way how the manuscript has been organized, or because this aspect is still difficult to fully link with confidence). Realistic suggestion: reorganizing the ideas would significantly improve the manuscript. In addition, interpretation of d13C is very complex compared with d18O, although the water-rock interaction may also complicate its interpretation. Among the factors that complicate the interpretation of the C records in speleothems is the-so called PCP (or prior calcite precipitation, or to be general Prior Carbonate Precipitation, to avoid discrimination between the two common CaCO3 polymorphs, calcite and aragonite). Could this factor influence the proxies being investigated in this study? E.g., for the large range (-8 to +1 per mil)?

Minor but crucial: There are some confusing technical terms used in the manuscript that need to be specified. For example, the word 'precipitation'. The authors should specify if the precipitation reflects *rainfall* which is climate or if it represents the *carbonate precipitation* leading to the formation of speleothems.

To summarize my general comment, I see that the dominant aspects of the hypotheses are focused on the local processes that may affect the carbon stable composition. The paper and the research are interesting, but there is plenty of room for

improvement. I hope my general comment and the detailed comments would help improving the paper.

**Detailed comments:**

L28- Please rephrase first sentence as: "Rapid and continuous analysis of radiocarbon concentration in carbonate samples at high resolution has been possible with the new LA-AMS technique."

L30: time

L42-43: I do not fully understand this mechanism, and it needs to be elaborated more

L47: I think there are some missing perspective remarks at the end of the abstract

L52: Fairchild et al. 2006: are you sure this is the best representative reference here?

L53: "except" Antarctica

L57: add a relevant literature reference after signal

L59: extra brackets (please remove, and anywhere in the manuscript)

L57-60: There is a recent paper by Fohlmeister et al. 2020, GCA that could be of relevance to this paragraph

L67: Some basics about dcf (as noted in the general comments), that could be an independent general context apart from the introduction

L72: "if a radiocarbon independent…" : why *if*?

L77: Values of dcf? (please specify)

L78: "commonly vary within a single speleothem with time" : reference please? Earlier you said no data still availabilities statement is therefore not proven nor supported by data?

L85: reference please?

L88: several studies? May be better to say two studies (as only two references are provided). Otherwise, please be accurate.

L89: dcf in what?

L90-92: I do not fully understand this statement, how can rainfall accelerate SOM decomposition?

how does rainfall increase the mean age of soil gas CO2?

L93: from some of the points in this paragraph, and looking back at the title, wouldn'it be more appropriate to use these two proxies for dissolution rather than climate?

Table 1: The context of these given values are not fully clear. Please explain in the table caption, and indicate under what type of vegetation cover?

L106: Please rewrite as "the studied stalagmite grew…." And please add the Lat/Long of the cave

L109: add a relevant reference for the geological aspect of the region

L134: what does "in situ" imply? , and please add a reference supporting that it was not ice covered during the Holocene (L135)

L160-163: aren't there be any markers in the stalagmite layers to match the wiggles? (please also seem more comments further below)

L218, 240: is it 14C or was there a typo?

L246: are displayed

Figure 4c: would be informative to see the age uncertainty by analysis, and the StalAge model with corresponding confidence interval. Where there is a discontinuity at 7.5ka?

L259: please add relevant reference as a guide for readers

L265-268: Please remove, if the structure of the manuscript is clear enough, this speech-kind of text can be easily removed. It brings no relevant information to understanding the significance of research.

L271: please refer to corresponding figures

L272: What bulk are you referring to?

L270–277: should belong to a short interpretation of the results. It does not give value to the title of the manuscript, which the authors should highlight the importance of F14C in paleoenvironmental reconstruction

General: This whole section 1 of the discussion can either belong to the method or part as a quick interpretation of the data, but it should not belong to the discussion section.

The authors should emphasize the importance of d13C and F14C in the use stalagmites in paleoenvironmental reconstruction

L310: epikarst (instead of karst), and please add a relevant reference at the end of the sentence at L311

L311-312: There are some contradictions here. More CacO3 dissolution does not mean nor imply higher growth rates. More dissolution may suggest more acidic solution, and thus supersaturation in CaCO3 was not achieved to allow calcite or aragonite to precipitate.

L313: please remove indeed

L314: Mangini et al. 2005: this reference is missing

L320: please correct for the typo. Also, do you mean here rainfall or CaCO3 precipitation?

L327: precipitation: please specify

L332: few re-writings suggestions: End the first sentence at "period." Then start with "The first is from soil CO2….. The second is from sulfuric acid….

L335: is it known how fast changes in soil d13C is transferred to speleothems?

L337-339: any references for this?

L340: why modern?

L344-345: this could be corrected prior to making comparison (see comment below)

L350: This is an instructive remark; however, it may be worth comparing (see comment below)

Comment about this paragraph: My problem with this part is that there is no correction done on the dft for d13C and 14C, but then the author commented on the correlation between the two variables. This aspect weakens the paper, because the authors could have done it better. For example, if the scanning speed of the laser is known (plus, they provide the spot size of the laser beam), it is possible to define the dft of the high/low peaks in the LA scans with the dft of the d13C. By selecting about 15–20 key points in the 14C data, I think it is possible to connect the 14C profile with the d13C profile with better accuracy.

L352-353: how about PCP?

L365: fast dripping =>fast growth: not necessarily. Fast drip sites in cave may be characterized by undersaturated solution with respect to CaCO3, and may not lead to CaCO3 precipitation (but instead, corrosion of the earlier deposited carbonates)

L370: low drip rate may in some cases increases the CaCO3 precipitation rates, unless the number of drops/sec is very slow, and hence could lead in hiatus in deposition

L378: Fractionation effects: please be precise, this title is quite vague

L381: lighter molecules change…. Please rephrase, it reads awkward

L382: Please delete : "this is valid for 13C and 14C isotopes". The statement does not bring any novelty

L386: any figures for reference here?

L388–390: the authors should elaborate on this with one or two sentences

L392: FigA7: this figure does not exist? Or did I miss it?

L398: long drip interval: I think if drip interval is long, this could enhance degassing/evaporation

L403: which is approximately -8permil (and please add reference)

L408: ref. please?

L428-431: here the authors clearly state that their approach could bring light on understanding processes affecting C isotopes in the subsurface (one of the reasons I made a comment about the mismatch between the title and the content of the manuscript)

L442: contributed $CO_2$ to $CaCO_3$: this is awkward, please rephrase

L463: why only $CO_2$ (shouldn't be DIC that is more appropriate?)

Figure 6 is very confusing to me; I am not fully sure what information to take from here? What is the importance of elevation in F14C?

L477: Re-writing suggestion:

"Results from this study allow to distinguish three intervals with different carbon dynamics.
  (1) The interval before 8ka BP….
  (2) The interval between 8–3.8ka BP (switch the number to follow the temporal logic)
  (3) ……"

L479: potentially enhancing bedrock dissolution.

---

## Author Comment (AC1) · 14 Mar 2021

**Responses to referee comments on "Climatic variations during the Holocene inferred from radiocarbon and stable carbon isotopes in a high-alpine cave"**

Our response is marked in blue.
We marked in yellow the questions of the referee.

**Referee #1:**

Key findings that I took away from this study: The coupled methodology (novel 14C measurements w/ stable isotopes) reveals changes of the presence of older OM reservoirs in the karst rock above the cave through time that are attributing to changing carbon isotope signals:
1. >8ka: there is an older reservoir of OM that causes low 13C and high dcf. *This interpretation wouldn't have been possible without the 14C measurements because low values of 13C alone lead to the interpretation of C3 vegetation above the cave and the measured high dcf is needed to distinguish this from the actual mechanism of vegetation + an older OM reservoirs source that is causing elevated CO2 values.

2. 3.8-8ka: There is strong δ13C variability and lower dcf. This suggests there is not an older source of OM contributing to the carbon isotope signal, and the interpretation is made it was stabilized/exhausted because of reduced precipitation. While this interpretation makes sense, I do not think bringing in growth rate is an accurate argument because I am not sure how "significant" a drop from 19µm to 30µm in growth rate is, and a change in growth rate could be from a variety of factors independent of precipitation amount (i.e. chemical kinetic processes, dissolved Ca2+ concentration, etc.). I recommend strengthening your argument on this (are their regional proxy records that suggest there was reduced precipitation amount at this time?) or removing the growth rate stance as a whole. Also, the interpretation is also made that "in-cave gas exchange processes are the most likely explanation for the strong 13C variability", and bedrock dissolution/fractionation processes are ruled out. There should be clarification for what you mean by "in-cave" gas exchange processes, because right now you make this interpretation, yet right before you rule out bedrock dissolution and fractionation mechanisms. *Also, why are the distinct isotopic changes at ⁓5 and 6ka not discussed? At both of these times, it appears 18O and dcf (⁓6ka), and 13C and dcf (⁓5ka) change drastically. Also at 6ka the dcf increases greatly – what could cause this? Why is this not mentioned? It definitely should be if the steady increase in dcf in the younger part of the sample is discussed (I find the sharp changes at 6-5 ka to be more significant than the younger part of the sample).
a) Growth rate and regional proxy records
- The drop in growth rate occurred from 50 to 30µm/a. This is a change by more then 40%, which is corroborated by several U-Th dates. In absolute numbers the change is not large, but relatively spoken it is a lot.
- We do agree that this change is likely not exclusively due to precipitation but could reflect reduced CaCO3 dissolution. We argue that the old deep OM reservoir had essentially been used up and less CO2 was available to acidify the solution, which could not dissolve CaCO3 anymore. Since we do not have additional records, which do show a reduction in prcp, we could indeed leave this away (or only speculate about it) but argue for a reduced CaCO3 dissolution due to a lower pCO2 in the skarst.
b) "In-cave" gas exchange processes: this is explained in line 396 – 398 in the manuscript: "Another process that may be dominant if the stalagmite growth rate is sufficiently low is C exchange between $CO_2$ of the cave air and C dissolved in the drip water."
c) We will describe this change more clearly in the revised manuscript, while admitting our ignorance about the processes leading to this. We will also stress that it remains enigmatic why d18O and dcf change at the same time at 6 ka but not at 5ka, while for d13C it is the temporal offset at 6ka, but a synchronous change at 5ka.

3. 2.4-3.8 ka: The more stable _13C signature and increase in dcf suggests a contri bution of aging OM reservoir. This old OM reservoir source is suggested to have been from the buildup of a mid-Holocene warm epoch. Overall, I follow this interpretation and thinks it makes sense. How/why is there the rapid decay trend of F14C at 6ka?

We will include the 6ka discussion at the appropriate section. However, we want to make sure at this point that the increase in dcf and decrease of f14c at 6 ka does not follow the 14C-decay trend. The change at 6 ka is much faster. Thus, we argue that there might be a change in the C source contributing to the rapid change. However, we have no explanation why this happens at this point. It may be related to the rapid decrease in d18O, but we are not sure.

**Comments:** Overall, I think this is a nice study and will make a worthwhile contribution to the literature. I think a future of combining novel high-resolution 14C measurements with other stable isotope data will definitely aid in the interpretation of these systems in terms of climactic processes. I think a bit of reworking with the discussion to help overall flow and clarifying several sections will result in a nice manuscript.

Some care should be taken when lumping speleothem "growth rate" with these interpretations – however, as a reduction in growth rate is not always a direct relation to reduced precipitation [i.e. changes in chemical kinetics, flow conditions (turbulent vs. laminar) could also control the growth rate of speleothems]. More of this is noted below, but this was one main issue I had with the interpretations.

I think this paper would strengthen if there was a brief, added section on what the initial 14C (F14C) raw values can tell us, vs. what the dcf tells us. The paper mentions what factors can contribute to dcf (lines 78-92), but it would be good to mention what the initial 14C values can tell us in this section as well. Especially because the F14C is referred to at the end of the paper (during the discussion of the youngest part of the sample, and in Fig. 6), which becomes slightly confusing because the entire paper is centered around the dcf values. I understand the distinction between the two (the use of initial 14C values to calculate for dcf) but adding a sentence to explain why you would look at raw, initial F14C values vs. dcf would be beneficial for overall flow and clarity of the manuscript and interpretations.

We will only use dcf in the new version of the manuscript.

In addition, there are lots of data in this manuscript and I'm confused what all your study specifically measured (the radiocarbon) vs. what data from other studies you are comparing it with (i.e. 18O, 13C? Did you all measure 13C?). I would make this clearer in the methods somewhere by stating directly: "in this study, we measured

This will be clarified at the end in the methods section.

Comments on tables & figures:

Table 1 – Why is there a "*" after the "expected" in the table. I see no explanation of this. Also, under the "Expected* 13C" column of the "Old OM contribution to seepage water acidification" row, does the "shift to more negative values" mean <0 or <10? If it's closer to <10, I'd label it like that since that's what you have in the above two rows (for consistency).

The "*" will be removed. The shift depends a bit on whether carbonic acid dissolution occurred in an open or closed system and whether sulfuric acid dissolution was involved. We will add: (< -10 permill possible)

Figure 1: There is no ruler in this photo. It says the length in the caption, but it would be helpful to have a ruler for reference. This would especially be helpful when looking back at this figure during the discussion when you say the "old section of SPA 127 (>8,ka, >120mm)", because then we'll be able to go and see where in the sample this is.

We will add a ruler to the Figure.

Figure 2: Are the d13C and d18O from other studies? If so, cite the studies in the figure caption after part (C).

d18O is from another study. D13C is so far unpublished but stems from the same measurements as d18O. However, the d13C interpretation was not possible at this time due to its complicated structure. We will add the citation.

Figure 2: Also, it would be helpful to plot the age in this figure (since age is what you refer to in the discussion, not depth). You could add it by an additional x-axis on the top.

We intentionally show the F14C against the distance from top in order to plot the raw data from this new method at the beginning. In Fig. 3 we then plot the dcf against the age and only this graph is used for interpretation.

Figure 3: You mention that "12CHE" is the signal intensity in the figure caption, but nowhere in the text do you explain this further. Can you add a section somewhere that says this? It becomes confusing during the discussion section (e.g. lines 270-273), because in these sections you refer to it only as "12C", and not 12CHE".

"HE" will be removed.

Figure 4: What stands out to me is the jump at 6ka (increase in dcf, decrease in _18O, and decrease in _13C), but this isn't mentioned in the text at all? How come?

See above. Of course we have noted this rapid and large change but have to admit, that we have no convincing explanation for this change. We will state it in a similar way in the new manuscript version (see also commet above).

Figure 4, overall comment: "the yellow, white, and orange shaded areas represent phases with distinct stable isotope and dcf characteristics" but what about the two sections (1: _6ka when dcf increases and dO18 and _13C decreases; 2: _5ka when _13C and dcf drops)? These transitions aren't really talked about in the text, and I'm wondering why you chose not to select these areas as "phases" with distinct characteristics?

Actually, we chose these growth periods because of the stable C isotope characteristics. We will remove "dcf" here.

Figure 4c: How do you know there are not hiatuses in between the U-Th ages? For example, it appears right before an age of 4 (the yellow/white boundary in panels 4a and 4b), there is a sharp decrease in dcf, increase in _18O), and increase in _13C), yet this is the part of your sample that has the longest gap of age control. How do you know there's not a micro-hiatus here that's undetectable? It may be worth mentioning you can't rule this hypothesis out, just to cover your bases and to let readers know you thought of this (rather than just interpreting this as a purely real signal). Also, the U-Th ages need error plotted.

We can exclude a long-lasting hiatus, even if the U-Th age determinations are sparse in this section. First, there is no macroscopic hint for this. In the whole section there is no distinct layer that potentially could point to a growth stop. Second, this part of SPA127 grew in parallel to speleothems SPA12 and SPA128, which where dated by ~ 10 points in this interval (Fohlmeister et al., 2013). None of the other two stalagmites shows a growth stop. The good correlation between the d18O signal of those three speleothems, thus, indicates that also no growth stop occurred in SPA127. However, we cannot exclude a microhiatus, although it appears very unlikely. But even if there is a microhiatus, this will have virtually no effect on the initial F14C or dcf. For example: an undected 'microhiatus' of 100 years (which is already quite long, but could be missed as derived by the small age errors and good match with d18O of other speleothems) will force f14C initial values to be off by 0.01 or in DCF by between 0.5 and 1 %. This is negligible compared to the change in DCF in this speleothem section. Thus we prefer not to discuss this topic at length in the revised manuscript, however, we will add U-Th error bars in Fig. 4 and state in section 'Materials and Methods' in subsection 'Sample' that hiati are very unlikely and provide the above explanation.

Comments by line:

Line 83-85: You mention the conditions in both an "open vs. closed" system can affect the dcf, but you only describe how the dcf typically is in an open system. I suggest adding a sentence describing what dcf would be in a closed system for clarity. Also detailing how there could be a change from an open to closed system may be helpful. I see you outline it in Figure 5, and also in Table 1, so perhaps just simply referencing these two figures/tables will help streamline this discussion.

We will change the manuscript accordingly.

Line 88-92: "Several studies show more closed-system conditions under higher precipitation regimes" maybe reword this sentence because I'm not sure what exactly you  mean by it. What constitutes "more closed-system" conditions? Once again, perhaps by referencing either Table 1 or Figure 5 this would help.

We will change the manuscript accordingly and explain better what is meant by this.

Line 136: How do you know there aren't hiatuses present in your speleothem? Perhaps  briefly state how you approached assessing hiatuses in your sample here. Also what  is the error on each age? I don't see this stated anywhere, and it's not in Figure 4C.  The error should be plotted.

See above.

Line 162-163: "offset between stable isotope and radiocarbon data of up to several  hundred micrometers", how did you go about accounting for this? Perhaps briefly describe  what you did to account for this offset so readers are aware of your methodology.  *I see in line 350 that you are unable to apply a correction factor. Perhaps stating this  earlier (such as at line 162-163) will help the reader better understand your process of  approaching this.

We will explain how we estimated this offset. It mainly arises from the spatial offset of the LA-AMS tracks to the stable isotope tracks. Growth layers are not linear and can be significantly distorted as we have seen from recent studies with our system. However, with our current LA cell, we cannot place the tracks closer to the edge of the sample. As suggested by the reviewer, we will also explain why we cannot apply a correction for both tracks (L162-163).

Line 193: It should be stated in the first sentence or two what you used this technique for. Example, directly stating: "FTIR was used for: : :.", because right now it is unclear  why you used FTIR (it's not until later in the discussion when you explain identifying  the contaminated epoyy area, and I think it'd be better to state up front in this section).

We will change the manuscript accordingly.

Line 276: "it's exact composition has been determined using FTIR". It is not until this  sentence that I realize what you are using FTIR for. Perhaps add a sentence to the  FTIR section (the section starting at line 193) that states, "we use FTIR to determine  specific compositions of areas in our sample to clarify the causes of anomalies."

See above.

Line 243: "For the more than 1500 radiocarbon data points", I suggest just inserting  the exact number of data analysis points that you have here, instead of saying "more  than.."

We will change the manuscript accordingly.

Line 259: Please add a reference at the end of this sentence to let readers know where  the "previously published 18O values" can be found. Also, did you measure the 13C  values in this study? Or did you pull data form another study? This is unclear and  should be clarified.

We state in the materials and methods section what we analyzed in this study. But as stated above, we will add this to the introduction.

Line 265: I think this paragraph could be reworded so it's clearer. I'm a bit confused  about how the different sections of the discussion are broken up the way they are. A suggestion: section "1. LA-AMS anomalies in the old section of : : :" should be an entirely  separate section than the ones below, because it just details how the presence  of epoxy caused contamination, and there are no other interpretations of the data as- sociated with it. This entire section be a brief paragraph at the start of the discussion  section, and then you could transition into a "part II" of the discussion that's exclusively  about the interpretation of the isotope systems across different parts of the sample.

We will move the epoxy discussion to the SI as suggested by Referee #2.

Line 266: A suggestion to clarify this sentence: "..in the bottom part of the sample"

We will change the manuscript accordingly.

Line 270: Please add the reference to Figure 3 after this sentence: "The five 12Ccururent  peaks correlating with.: : :(Figure 3)" so the readers know this is what you're  referencing. Also change "indicating" to "indicate".

We will change the manuscript accordingly.

Lines 300, 328, 335, 337, 339, 396, 397: Change "C" to "C-isotope". I probably missed  additional places you refer to just "C", so please change everywhere this occurs.

We will change the manuscript accordingly.

Comment on section "2. Old section of SPA (>8 ka BP) (Line 293-315): The first paragraph of this section (line 294-309) walks readers through interpretations of high dcf values and low 13C values from >8ka. A transitional sentence is needed in the beginning part of the second paragraph (lines 310-315), to set up the connection of the two (i.e. the warmer temperatures for the Holocene could have caused the mobilization of the older OM), because right now it feels out of place a bit. Also adding a concluding sentence would be beneficial to wrap up this section of the discussion.

We will change the manuscript accordingly.

Line 319: Perhaps state at what depth/age you are referring to here in this sentence. For example: "As indicated by the reduced growth rate in SPA 127 (Fig. 4, 3.8 ka).."

We will change the manuscript accordingly.

Line 320: Misspell of "precipitation" (it says "recipitatoin").

We will change the manuscript accordingly.

Line 321: Please add this to the sentence for clarity: "The low 13C -values of the first growth period are superseded by rapid and very large variations of 13C."

We will change the manuscript accordingly.

Line 325-328: You state here "the dcf between 3.8 and 8 ka is generally lower than the older section: but at ⬚5.8-6ka dcf jumps relatively high and stays high until ⬚5.2 ka. I think this should be addressed somewhere in your discussion.

We will add this to the discussion, although we cannot explain what might have caused this change (see above).

Line 343, 392: what is (Fig A7?) Do you mean supplemental Figure 7 (Fig S7)?

Yes. This will be changed accordingly.

Line 330: Hypothesis 1, general comments –

Line 351: "a positive correlation between main features of 13C and dcf are observable for the middle period, especially between 3.8-5 ka and 6-8 ka BP." I would argue @ 4ka they appear reversed, but 6-8ka I believe I see this correlation. A suggestion: zoom in on these two time slices for what's plotted in Figure 4B, especially the 6-8 ka one, so the correlation is clearer (the blue and orange lines are kind of on top of each other in the figure now so it's hard to see). The argument for this could be stronger if you could demonstrate the relationship more clearly.

We well add such a zoom in the Figure.

Line 354: A bit more explanation for why "an increase of the dcf to 100%" is needed for this mechanism to work would help the flow of this argument better.

Because there wouldn't be any modern 14C signature from the soil, only old bedrock carbon. We will add a sentence here.

Line 355: "Generally, the dcf is even smaller than in the youngest and oldest section of the stalagmite.." What about at from 6-5ka? This need addressed.

See above.

Line 354: "..this is expected to be accompanied by an increase of the dcf to 100%." Why? Some elaboration on this would strengthen your argument.

Will be added. See above.

Line 362: Hypothesis 2, general comments – Overall the text in the discussion of this hypothesis is clear. But I disagree with your growth rate argument. As stated in previous comments, I'm not sure if a change in the growth rate (19⬚m/11⬚m to 30⬚m ) is "significant" – I consider this just a "lower" growth rate. I therefore don't think you can use this piece of information to suggest it was caused by an overall reduction in precipitation amount. I think you should either try to bring in literature that demonstrate regional drier conditions, or some other support for this argument other than growth rate.

We will replace significant by a relative number of change e.g. 30 to 60% reduction in growth rate. We agree that a reduction in growth rate is not the only potential reason and there are multiple ways to interpret this signal: 1) less prescipitation and 2) an exhausted OM reservoir. We will discuss both options because we think, a reduction in prcp could be very well responsible for the decrease in growth rate. Alternatively, there was reduced contribution of the decreased OM to the acidification

of the solution in the karst, which resulted in lower Ca2+ concentrations. This has the same effect as a lower growth rate. We will modify this hypothesis in the next version.

Lines 395-427: I'm a bit confused with this paragraph. I follow your discussion, but are you saying this is your main interpretation for what is happening during this interval and causing all the fluctuations? (i.e. is this what you mean by "in-cave" processes). You state in the conclusion that it's not bedrock dissolution or fractionation processes, so are you interpreting the strong variability in 13C as a change of gas exchange processes? If so, this needs to be stated clearer, because right now it's a bit ambiguous whether you mean this or not.

Yes, this is exactly what we mean and we will state this in a clearer way.

---

## Author Comment (AC2) · 14 Mar 2021

Responses to referee comments on "Climatic variations during the Holocene inferred from radiocarbon and stable carbon isotopes in a high-alpine cave"

Our response is marked in blue.
We marked in ==yellow== the questions of the referee.

**Referee #2:**

The strength of the paper is the combination of radiocarbon concentration and d13 C in stalagmites samples at a continuous and high spatial resolution to understand what happens above the Spannagel cave during some period of the Holocene. However, I found that the manuscript is moderately written (please don't take this personal, and as a reviewer I am not qualified to comment in detail about the writing, but I strongly believe there is a room for improvement and writing is a learning process).

Also, in my opinion, the manuscript requires a lot of reorganization of ideas to make it clearer. For example, the first section of the discussion section, the anomalies in the old section of SPA 127 do not highlight the importance of the proxies being studied. Instead, it discusses the result based on methodological approach. Please note that the journal "Climate of the Past" is not a methodology journal (if this paper was submitted to a method paper, then I would not argue about having this part in the discussion). In fact, I see that this whole section either belongs to the method or some part of it could go as a quick interpretation of the data in the result section, but it should not belong to the discussion section.

We agree and we will follow the advice of the referee to move this part from the discussion section and only shortly describe it in the results and in the SI. Additionally, we will move Figure 2 and 3 in the SI and instead add a new zoom-in Fig. of current Fig.4 as suggested by Referee 1

Another example, a separate generality section about dcf can be helpful here (see detailed comments) where the authors could explain the difference between radioactive 14C and dead carbon, and what are the potential sources of them. With this said, the sections at L 67-105 could belong to that ==independent section== right between the introduction and the Materials &methods. If that general&fundamental notion is separated from the introduction, I am certain that the introduction could become concise and clearer, with a clear statement of the problematic, and a proposition of the new method and its potential relevance in future paleo– reconstruction.

As stated above, we will only use the dcf and not the initial 14C.

I also feel the title does not fully capture the content of the manuscript. While the authors entitled their manuscript "Climatic variations during the Holocene inferred from radiocarbon and stable carbon isotopes in a high Alpine cave", I found that the manuscript mainly use radiocarbon and d 13 C as a proxy for local changes and specifically what happens right above the cave in the epikarst, and not directly to climate. In their conclusion, it was made clear that these two are good proxies to understand carbon dynamic. ==Hence, I think the authors should emphasize the importance of d13C and F14C in the use of stalagmites in paleoenrivonmental reconstruction and build their discussion based on that, rather than jumping directly to climate, which at this stage seems more speculative.==

-There are several points by the authors in manuscript that support my comments. For example, in the abstract, the authors used the variation in 14C and d13C as an evidence of host bedrock dissolution or organic matter reservoir contribution from the epikarst to the cave. And in fact, this has been one of the focuses of the interpretation/discussion. ==The authors should make that clear that from using such inferences,== information from the local place can later be applied to climatic context.

Same as stated above-

-In my reading of the manuscript, the bridge 'local response–climate' is quite obscure (possibly by the current way how the manuscript has been organized, or because this aspect is still difficult to fully link with confidence). Realistic suggestion: ==reorganizing the ideas would significantly improve the manuscript.==

We will add a paragraph to the introduction (see above)

-In addition, interpretation of d13C is very complex compared with d18O, although the water-rock interaction may also complicate its interpretation. Among the factors that complicate the interpretation of the C records in speleothems is the-so called PCP (or prior calcite precipitation, or to be general Prior Carbonate Precipitation, to avoid discrimination between the two common CaCO3 polymorphs, calcite and aragonite). Could this factor influence the proxies being investigated in this study? E.g., for the large range (-8 to +1 per mil)?

We agree and we will add a third paragraph (iii) under hypothesis 2 and name it PCP. PCP can have an effect on d13C, even as large ones as observed for our stalagmite. While this would not have an effect on 14C, we would expect, that d18O should show a similar behavior, which is not the case. Thus, we can savely assume, that PCP is not responsible.

Minor but crucial: There are some confusing technical terms used in the manuscript that need to be specified. For example, the word 'precipitation'. The authors should specify if the precipitation reflects rainfall which is climate or if it represents the carbonate precipitation leading to the formation of speleothems.

We will clarify this throughout the manuscript.

To summarize my general comment, I see that the dominant aspects of the hypotheses are focused on the local processes that may affect the carbon stable composition. The paper and the research are interesting, but there is plenty of room for improvement. I hope my general comment and the detailed comments would help improving the paper.

We will also address the detailed comments.

---

## Author Response (AR1)

**Documentation of changes to our manuscript:**

**Climatic variations during the Holocene inferred from radiocarbon and stable carbon isotopes in a high-alpine cave**

Caroline Welte[1,2], Jens Fohlmeister[3,4], Melina Wertnik[1,2], Lukas Wacker[1], Bodo Hattendorf[5], Timothy I. Eglinton[2], Christoph Spötl[6]

Our answers are written in green. If not stated differently, we refer to line numbers of the original manuscript.

General changes in accordance with comments made by both referees regarding reorganization of the manuscript:

1. Everything related to the epoxy/14C anomalies is moved to the SI
2. Line 226 – 237, Fig. 2 and Fig. 3 moved to SI.
3. We created a new section (Section 2) in which we explain the use of 14C in speleothems, define the term dead carbon fraction and how it is linked to climate.
4. Throughout the manuscript, the radiocarbon concentration is now only expressed as dcf. Therefore, section 1 under results "radiocarbon" is moved to the SI. The new section 2 under Results is renamed to "Radiocarbon (dcf)".

**Referee#1: Comments on tables & figures:**

Table 1 – Why is there a "*" after the "expected" in the table. I see no explanation of this. Also, under the "Expected* 13C" column of the "Old OM contribution to seepage water acidification" row, does the "shift to more negative values" mean <0 or <10? If it's closer to <10, I'd label it like that since that's what you have in the above two rows (for consistency).
Asterisk removed, "< -10‰" added instead of "shift towards more negative values"

Figure 1: There is no ruler in this photo. It says the length in the caption, but it would be helpful to have a ruler for reference. This would especially be helpful when looking back at this figure during the discussion when you say the "old section of SPA 127 (>8,ka, >120mm)", because then we'll be able to go and see where in the sample this is.
1. ruler is now added and green box is removed (which indicated the region of 14C anomalies). The original Fig. 1 is now moved to the SI.
2. Caption of Fig. 1 is now shortened: "Picture of SPA 127 (top is left). Top and bottom piece with location of the stable isotope track marked in purple and LA-AMS test track in blue (the tracks corresponding to the data presented in this work were placed next to the test tracks). The total length of the slab is 14.6 cm. "

Figure 2: Are the d13C and d18O from other studies? If so, cite the studies in the figure caption after part (C).
1. The reference for the d18O measurements is added in the caption of Fig 4 (now Fig 2).

2. Line 120: added "measured in this study"
3. Line 121: added "previously published"

Figure 2: Also, it would be helpful to plot the age in this figure (since age is what you refer to in the discussion, not depth). You could add it by an additional x-axis on the top.
Fig. 2 is now moved to the SI.

Figure 3: You mention that "12CHE" is the signal intensity in the figure caption, but nowhere in the text do you explain this further. Can you add a section somewhere that says this? It becomes confusing during the discussion section (e.g. lines 270-273), because in these sections you refer to it only as "12C", and not 12CHE".
Fig. 2 is now moved to the SI and "HE" is removed.

Figure 4: What stands out to me is the jump at 6ka (increase in dcf, decrease in _18O, and decrease in _13C), but this isn't mentioned in the text at all? How come?

See further down.

Figure 4, overall comment: "the yellow, white, and orange shaded areas represent phases with distinct stable isotope and dcf characteristics" but what about the two sections (1: _6ka when dcf increases and dO18 and _13C decreases; 2: _5ka when _13C and dcf drops)? These transitions aren't really talked about in the text, and I'm wondering why you chose not to select these areas as "phases" with distinct characteristics?
Actually, we chose these growth periods because of the stable C isotope characteristics. We removed "dcf" here. As we are not able to explain the dcf change at 6 and 5 ka (see above) we refrain from marking this interval as well.

Figure 4c: How do you know there are not hiatuses in between the U-Th ages? For example, it appears right before an age of 4 (the yellow/white boundary in panels 4a and 4b), there is a sharp decrease in dcf, increase in _18O), and increase in _13C), yet this is the part of your sample that has the longest gap of age control. How do you know there's not a micro-hiatus here that's undetectable? It may be worth mentioning you can't rule this hypothesis out, just to cover your bases and to let readers know you thought of this (rather than just interpreting this as a purely real signal). Also, the U-Th ages need error plotted.
L136: we added in the text: "There is no macro- and microscopic evidence for the existence of hiatuses in this specimen. Further evidence for the absence of hiatuses is provided by two additional speleothems, SPA 12 and SPA 128, from the same cave are partly coeval with SPA 127. These additional speleothems have a higher dating density in parts, where SPA 127 has only a few radiometric U-Th dating points and also do not show evidence of hiatuses (Fohlmeister et al., 2013). In combination with the well replicated stable O isotope signals we are confident that the growth of SPA 127 was not affected by hiatuses."

The sign size is larger than the U-Th errors. We added this information in the previous Fig. 4 (now Fig. 2) caption. We added StalAge-derived upper and lower limits of the age-depth model.

Comments by line:

Line 83-85: You mention the conditions in both an "open vs. closed" system can affect the dcf, but you only describe how the dcf typically is in an open system. I suggest adding a sentence describing what dcf would be in a closed system for clarity. Also detailing how there could be a change from an open to closed system may be helpful. I see you outline it in Figure 5, and also in Table 1, so perhaps just simply referencing these two figures/tables will help streamline this discussion.

Line 85: added "In a more closed system, this exchange is inhibited with the extreme case being a completely closed system, where for each mole of carbonic acid one mole of CaCO3 is dissolved resulting in a dcf of up to 50%."

Line 88-92: "Several studies show more closed-system conditions under higher precipitation regimes" maybe reword this sentence because I'm not sure what exactly you mean by it. What constitutes "more closed-system" conditions? Once again, perhaps by referencing either Table 1 or Figure 5 this would help.
Line 91: added "(see Table 1)» and see point before.

Line 136: How do you know there aren't hiatuses present in your speleothem? Perhaps briefly state how you approached assessing hiatuses in your sample here. In addition, what is the error on each age? I don't see this stated anywhere, and it's not in Figure 4C. The error should be plotted.
See above.

Line 162-163: "offset between stable isotope and radiocarbon data of up to several hundred micrometers", how did you go about accounting for this? Perhaps briefly describe what you did to account for this offset so readers are aware of your methodology. *I see in line 350 that you are unable to apply a correction factor. Perhaps stating this earlier (such as at line 162-163) will help the reader better understand your process of approaching this.
We moved the description from L350 to L162 and rephrased it to: "LA-scans were placed as close as possible to the stable isotope tracks in order to facilitate matching between the two data sets (Fig. 2). However, the LA-AMS setup does not permit to place laser tracks close to the rim of samples causing an offset between the two sampling lanes of approximately 5 mm. Speleothem growth layers are often curved, resulting in a potential offset between stable isotope and radiocarbon data of up to several hundred micrometers, with the outer LA-scan appearing somewhat older than the stable isotope record. Since the curvature of the growth layers is most likely variable, a constant correction factor has not been applied."

L350: rephrased to "Taking a potential offset between the two records into account".

Line 193: It should be stated in the first sentence or two what you used this technique for. Example, directly stating: "FTIR was used for: : :.", because right now it is unclear why you used FTIR (it's not until later in the discussion when you explain identifying the contaminated epoyy area, and I think it'd be better to state up front in this section).
Line 194: we added "Fourier Transform Infrared spectroscopy (FTIR) was used to determine the specific composition of selected areas in our sample to clarify the causes of anomalies." (This part has been moved to SI now).

Line 276: "it's exact composition has been determined using FTIR". It is not until this sentence that I realize what you are using FTIR for. Perhaps add a sentence to the FTIR section (the section starting at line 193) that states, "we use FTIR to determine specific compositions of areas in our sample to clarify the causes of anomalies."
See above.

Line 243: "For the more than 1500 radiocarbon data points", I suggest just inserting the exact number of data analysis points that you have here, instead of saying "more than.."
Now stated the exact number: 1402

Line 259: Please add a reference at the end of this sentence to let readers know where the "previously published 18O values" can be found. Also, did you measure the 13C values in this study? Or did you pull data form another study? This is unclear and should be clarified.

See above

Line 265: I think this paragraph could be reworded so it's clearer. I'm a bit confused about how the different sections of the discussion are broken up the way they are. A suggestion: section "1. LA-AMS anomalies in the old section of:" should be an entirely separate section than the ones below, because it just details how the presence of epoxy caused contamination, and there are no other interpretations of the data as- sociated with it. This entire section be a brief paragraph at the start of the discussion section, and then you could transition into a "part II" of the discussion that's exclusively about the interpretation of the isotope systems across different parts of the sample.
We moved the epoxy discussion to the SI as suggested by Referee #2 and changed the introductory sentences of the Discussion (L265-268) accordingly.

Line 266: A suggestion to clarify this sentence: "..in the bottom part of the sample"
Manuscript changed accordingly.

Line 270: Please add the reference to Figure 3 after this sentence: "The five 12Ccururent peaks correlating with.: : :(Figure 3)" so the readers know this is what you're referencing. Also change "indicating" to "indicate".
This part is now moved to the SI.

Lines 300, 328, 335, 337, 339, 396, 397: Change "C" to "C-isotope". I probably missed additional places you refer to just "C", so please change everywhere this occurs.
Manuscript changed accordingly.

Comment on section "2. Old section of SPA (>8 ka BP) (Line 293-315): The first paragraph of this section (line 294-309) walks readers through interpretations of high dcf values and low 13C values from >8ka. A transitional sentence is needed in the beginning part of the second paragraph (lines 310-315), to set up the connection of the two (i.e. the warmer temperatures for the Holocene could have caused the mobilization of the older OM), because right now it feels out of place a bit. Also adding a concluding sentence would be beneficial to wrap up this section of the discussion.
We changed the manuscript accordingly.

Line 319: Perhaps state at what depth/age you are referring to here in this sentence. For example: "As indicated by the reduced growth rate in SPA 127 (Fig. 4, 3.8 ka).."
Manuscript changed accordingly.

Line 320: Misspell of "precipitation" (it says "recipitatoin").
Manuscript changed accordingly.

Line 321: Please add this to the sentence for clarity: "The low 13C -values of the first growth period are superseded by rapid and very large variations of 13C."
Manuscript changed accordingly.

Line 325-328: You state here "the dcf between 3.8 and 8 ka is generally lower than the older section: but at 5.8-6ka dcf jumps relatively high and stays high until 5.2 ka. I think this should be addressed somewhere in your discussion.
This is now added to the discussion (L249-254 in the revised manuscript).

Line 343, 392: what is (Fig A7?) Do you mean supplemental Figure 7 (Fig S7)?
Manuscript changed accordingly.

Line 330: Hypothesis 1, general comments –

Line 351: "a positive correlation between main features of 13C and dcf are observable for the middle period, especially between 3.8-5 ka and 6-8 ka BP." I would argue @ 4ka they appear reversed, but 6-8ka I believe I see this correlation. A suggestion: zoom in on these two time slices for what's plotted in Figure 4B, especially the 6-8 ka one, so the correlation is clearer (the blue and orange lines are kind of on top of each other in the figure now so it's hard to see). The argument for this could be stronger if you could demonstrate the relationship more clearly.

Enlargements are now provided. We also differentiated between both periods, as the 6-8 ka period shows a better correlation than in the 3.8 – 5 ka period.

Line 354: A bit more explanation for why "an increase of the dcf to 100%" is needed for this mechanism to work would help the flow of this argument better.

Added: "because of the diminishing $^{14}$C-rich soil signature"

AND
Line 355: "Generally, the dcf is even smaller than in the youngest and oldest section of the stalagmite.." What about at from 6-5ka? This need addressed.

See above.

Line 354: "..this is expected to be accompanied by an increase of the dcf to 100%." Why? Some elaboration on this would strengthen your argument.

See above.

Line 362: Hypothesis 2, general comments – Overall the text in the discussion of this hypothesis is clear. But I disagree with your growth rate argument. As stated in previous comments, I'm not sure if a change in the growth rate (19m/11m to 30m ) is "significant" – I consider this just a "lower" growth rate. I therefore don't think you can use this piece of information to suggest it was caused by an overall reduction in precipitation amount. I think you should either try to bring in literature that demonstrate regional drier conditions, or some other support for this argument other than growth rate.

We agree that a reduction in growth rate is not the only potential reason and there are multiple ways to interpret this signal: 1) less precipitation and 2) an exhausted OM reservoir. We discuss both options in the beginning of section 2 in the discussion, because we think, a reduction in precipitation could be very well responsible for the decrease in growth rate. Alternatively, there was reduced contribution of the decreased OM to the acidification of the solution in the karst, which resulted in lower Ca2+ concentrations. This has the same effect as a lower growth rate. We will modified this in the current manuscript.

Lines 395-427: I'm a bit confused with this paragraph. I follow your discussion, but are you saying this is your main interpretation for what is happening during this interval and causing all the fluctuations? (i.e. is this what you mean by "in-cave" processes). You state in the conclusion that it's not bedrock dissolution or fractionation processes, so are you interpreting the strong variability in 13C as a change of gas exchange processes? If so, this needs to be stated clearer, because right now it's a bit ambiguous whether you mean this or not.

L396-399 We rephrased this and attempt to be clearer: "Another process that is a potential candidate for causing the behavior observed in SPA 127 in this interval is the C isotope exchange between CO2 of the cave air and C dissolved in the drip water. The gas exchange process may be dominant if the stalagmite growth rate is sufficiently low and when drip intervals are long and/or the differences between the pCO2 of the water and cave air is small ".

**Referee #2**

The strength of the paper is the combination of radiocarbon concentration and d13 C in stalagmites samples at a continuous and high spatial resolution to understand what happens above the Spannagel cave during some period of the Holocene. However, I found that the manuscript is moderately written (please don't take this personal, and as a reviewer I am not qualified to comment in detail about the writing, but I strongly believe there is a room for improvement and writing is a learning process).

Also, in my opinion, the manuscript requires a lot of reorganization of ideas to make it clearer. For example, the first section of the discussion section, the anomalies in the old section of SPA 127 do not highlight the importance of the proxies being studied. Instead, it discusses the result based on methodological approach. Please note that the journal "Climate of the Past" is not a methodology journal (if this paper was submitted to a method paper, then I would not argue about having this part in the discussion). In fact, I see that this whole section either belongs to the method or some part of it could go as a quick interpretation of the data in the result section, but it should not belong to the discussion section.

We moved this part from the discussion section to the SI and only shortly describe it in the results. Additionally, we moved Figure 2 and 3 in the SI and instead add a new zoom-in Fig. of current Fig.4 as suggested by Referee 1.

Another example, a separate generality section about dcf can be helpful here (see detailed comments) where the authors could explain the difference between radioactive 14C and dead carbon, and what are the potential sources of them. With this said, the sections at L 67-105 could belong to that independent section right between the introduction and the Materials &methods. If that general&fundamental notion is separated from the introduction, I am certain that the introduction could become concise and clearer, with a clear statement of the problematic, and a proposition of the new method and its potential relevance in future paleo– reconstruction.

We now only use the dcf and not the initial 14C. This has also been changed in Fig. 6 (including the caption). L 67-105 are now moved to a new section right after the introduction named "radiocarbon and dead carbon fraction" that aims at clarifying the difference between the two and discusses the sources and processes affecting the dcf. Here, we added a new Figure (Figure 1) that shows the typical C signatures in karst environments.

I also feel the title does not fully capture the content of the manuscript. While the authors entitled their manuscript "Climatic variations during the Holocene inferred from radiocarbon and stable carbon isotopes in a high Alpine cave", I found that the manuscript mainly use radiocarbon and d 13 C as a proxy for local changes and specifically what happens right above the cave in the epikarst, and not directly to climate. In their conclusion, it was made clear that these two are good proxies to understand carbon dynamic. Hence, I think the authors should emphasize the importance of d13C and F14C in the use of stalagmites in paleoenrivonmental reconstruction and build their discussion based on that, rather than jumping directly to climate, which at this stage seems more speculative.

- There are several points by the authors in manuscript that support my comments. For example, in the abstract, the authors used the variation in 14C and d13C as an evidence of host bedrock dissolution or organic matter reservoir contribution from the epikarst to the cave. And in fact, this has been one of the focuses of the interpretation/discussion. The authors should make that clear that from using such inferences, information from the local place can later be applied to climatic context.
  We explain how dcf and d13C can be linked to climate in section 2: Radiocarbon and dead carbon fraction

- In my reading of the manuscript, the bridge 'local response–climate' is quite obscure (possibly by the current way how the manuscript has been organized, or because this aspect is still difficult to fully link with confidence). Realistic suggestion: reorganizing the ideas would significantly improve the manuscript.

  Yes, it is still difficult to fully link changes in C-isotopes to climate variations. But in our work we were able to provide evidence for climate related changes in the various proxies of SPA127.
  We also follow the reviewers ideas for restructuring the ms in order to make the points clearer.

  In addition, interpretation of d13C is very complex compared with d18O, although the water-rock interaction may also complicate its interpretation. Among the factors that complicate the interpretation of the C records in speleothems is the-so called PCP (or prior calcite precipitation, or to be general Prior Carbonate Precipitation, to avoid discrimination between the two common CaCO3 polymorphs, calcite and aragonite). Could this factor influence the proxies being investigated in this study? E.g., for the large range (-8 to +1 per mil)?

  We agree that this might be an option for the rapid changes in d13C. Therefore we added a third paragraph under hypothesis 2 and named it PCP and added an explanation, why this cannot be responsible for the observed signal. PCP can have an effect on d13C, even as large as observed for our stalagmite. While this would not have an effect on 14C, we would expect, that d18O should show a similar behavior, which is not the case. Thus, we can safely assume, that PCP is not responsible.

Minor but crucial: There are some confusing technical terms used in the manuscript that need to be specified. For example, the word 'precipitation'. The authors should specify if the precipitation reflects rainfall which is climate or if it represents the carbonate precipitation leading to the formation of speleothems.

We clarified this throughout the manuscript.

To summarize my general comment, I see that the dominant aspects of the hypotheses are focused on the local processes that may affect the carbon stable composition. The paper and the research are interesting, but there is plenty of room for improvement. I hope my general comment and the detailed comments would help improving the paper.

Detailed comments:

L28- Please rephrase first sentence as: "Rapid and continuous analysis of radiocarbon concentration in carbonate samples at high resolution has been possible with the new LAAMS technique."
Done and we added "very"
L30: time
removed
L42-43: I do not fully understand this mechanism, and it needs to be elaborated more
This is now explained in detail in the discussion section (Hypothesis II, part III, line 326 of revised manuscript).
L47: I think there are some missing perspective remarks at the end of the abstract
Added: This study reveals the high potential of combining high-resolution 14C profiles in speleothems with δ13C records in order to disentangle climate-related C dynamics in karst systems.
L52: Fairchild et al. 2006: are you sure this is the best representative reference here?
We added other, more appropriate references.
L53: "except" Antarctica
Done
L57: add a relevant literature reference after signal

Done

L59: extra brackets (please remove, and anywhere in the manuscript)

Done

L57-60: There is a recent paper by Fohlmeister et al. 2020, GCA that could be of relevance to this paragraph

Added

L67: Some basics about dcf (as noted in the general comments), that could be an independent general context apart from the introduction

This has its own section now and is described in more detail.

L72: "if a radiocarbon independent…" : why if?

Because otherwise the dcf cannot be calculated. Either because there is no age-depth model at all or if the chronology was established by radiocarbon dates, an assumption for the dcf had to be made to establish an age-depth model.

L77: Values of dcf? (please specify)

done

L78: "commonly vary within a single speleothem with time" : reference please? Earlier you said no data still availabilities statement is therefore not proven nor supported by data?

We added references here.

L85: reference please?

We added a reference.

L88: several studies? May be better to say two studies (as only two references are provided). Otherwise, please be accurate.

There are more studies available than only two, which show such a behavior. We added three more references.

L89: dcf in what?

Added: in the stalagmite

L90-92: I do not fully understand this statement, how can rainfall accelerate SOM decomposition? how does rainfall increase the mean age of soil gas CO2?

We removed this.

L93: from some of the points in this paragraph, and looking back at the title, wouldn'it be more appropriate to use these two proxies for dissolution rather than climate?

Yes, indeed it is possible to derive dissolution conditions from both proxies. However, dissolution conditions have been shown to depend on climate (e.g., Griffiths et al. 2012; Noronha et al., 2014; Bajo et al., 2017; Therre et al., 2020)

Table 1: The context of these given values are not fully clear. Please explain in the table caption, and indicate under what type of vegetation cover?

We improved the table according to suggestions by Reviewer 1 and added that the d13C values are expected under a C3 vegetation cover.

L106: Please rewrite as "the studied stalagmite grew…." And please add the Lat/Long of the cave

Done in Line 127 (mat & Method section)

L109: add a relevant reference for the geological aspect of the region

Added: Spötl et al. 2004

L134: what does "in situ" imply? , and please add a reference supporting that it was not ice covered during the Holocene (L135)

We removed in-situ and added Fohlmeister et al., 2013 as a reference.

L160-163: aren't there be any markers in the stalagmite layers to match the wiggles? (please also seem more comments further below)

We provided now more details, why it is not possible to correct the measurement depths offsets from the stable isotope and 14C sampling tracks (Lines 160-165 in the revised manuscript)

L218, 240: is it 14C or was there a typo?

Yes, it the 14C measurements we are referring to in both lines. It is no typo.

L246: are displayed

Done

Figure 4c: would be informative to see the age uncertainty by analysis, and the StalAge model with corresponding confidence interval. Where there is a discontinuity at 7.5ka?

Changed accordingly. No at 7.5ka is not discontinuity. Isotope values are missing there. Therefore, there is a gap. We improved this now by closing the gap in the updated figure.

L259: please add relevant reference as a guide for readers

Done

L265-268: Please remove, if the structure of the manuscript is clear enough, this speechkind of text can be easily removed. It brings no relevant information to understanding the significance of research.

Shortened to: The interpretation of the results on C isotopes in SPA 127 will be divided in three main parts that correspond to three sections identified on the speleothem through distinct $\delta^{13}C$ characteristics.

L271: please refer to corresponding figures

Moved to SI.

L272: What bulk are you referring to?

Moved to SI.

L270–277: should belong to a short interpretation of the results. It does not give value to the title of the manuscript, which the authors should highlight the importance of F14C in paleoenvironmental reconstruction

General: This whole section 1 of the discussion can either belong to the method or part as a quick interpretation of the data, but it should not belong to the discussion section. The authors should emphasize the importance of d13C and F14C in the use stalagmites in paleoenvironmental reconstruction

We transferred the whole section of the LA-AMS anomalies to the SI.

L310: epikarst (instead of karst), and please add a relevant reference at the end of the sentence at L311

Changed to epikarst. We added "likely" here.

L311-312: There are some contradictions here. More CacO3 dissolution does not mean nor imply higher growth rates. More dissolution may suggest more acidic solution, and thus supersaturation in CaCO3 was not achieved to allow calcite or aragonite to precipitate.

Here, we disagree with the reviewer. Usually carbonate dissolution is a fast process, which happens in the range of hours (Dreybrodt and Scholz, 2011) until saturation is achieved. But even if saturation with respect to Ca2+ should not be achieved, a higher soil air pCO2 leads to a faster dissolution, resulting in a larger degree of CaCO3 dissolution and in a higher Ca2+ concentration. Once such Ca2+ - undersaturated solutions enter the cave, they become pretty fast saturated and supersaturated with respect to Ca2+, as CO2 will degas immediately, as cave air CO2 values are smaller than that of the soil gas. Then the water, which had the change to dissolve more CaCO3, will potentially be able to precipitate more CaCO3.

L313: please remove indeed

Done

L314: Mangini et al. 2005: this reference is missing

Done

L320: please correct for the typo. Also, do you mean here rainfall or CaCO3 precipitation?

Done

L327: precipitation: please specify

Done

L332: few re-writings suggestions: End the first sentence at "period." Then start with "The first is from soil CO2….. The second is from sulfuric acid….

Done

L335: is it known how fast changes in soil d13C is transferred to speleothems?

That is a difficult question. We would think that it is not possible to see any change in soil gas d13C earlier than the mean water residence time, which is typically in range of years for most caves. But within a few decades such soil gas d13C change should be visible. But there is no study around, which investigates such a change systematically. Therefore, we do not change the text with respect to this question.

L337-339: any references for this?

We added Bajo et al. (2017) and Spötl et al. (2016)

L340: why modern?

Added "comparably"

L344-345: this could be corrected prior to making comparison (see comment below)

See below.

L350: This is an instructive remark; however, it may be worth comparing (see comment below) Comment about this paragraph: My problem with this part is that there is no correction done on the dft for d13C and 14C, but then the author commented on the correlation between the two variables. This aspect weakens the paper, because the authors could have done it better. For example, if the scanning speed of the laser is known (plus, they provide the spot size of the laser beam), it is possible to define the dft of the high/low peaks in the LA scans with the dft of the d13C. By selecting about 15–20 key points in the 14C data, I think it is possible to connect the 14C profile with the d13C profile with better accuracy.

We agree with the reviewer. Usually, multi proxy studies with speleothems often suffer from missing alignment between individual measurement tracks. However, often this problem is not mentioned in manuscripts. We appreciate and agree with the suggestion of the reviewer for any improvement. Unfortunately, it is not a problem of not knowing the distance from top of the LA-track. The problem lies in the curvature of individual growth layers, which might be different for different parts of the stalagmite. Thus, even a spatial distance of only 0.5 cm between the stable isotope and LA-tracks can cause some problems in alignment. In order to make this problem more understandable, we moved the description from L350 to L162 and rephrased it to a more detailed description of the problem: "LA-scans were placed as close as possible to the stable isotope tracks in order to facilitate matching between the two data sets (Fig. 1). However, the LA-AMS setup does not permit to place laser tracks close to the rim of samples causing a distance between the two sampling lanes of approximately 5 mm. Speleothem layers are often curved, resulting in a potential offset between stable isotope and radiocarbon data of up to several hundred micrometers, with the outer LA-scan appearing somewhat older than the stable isotope record. Since the curvature of the growth layers is most likely variable, a constant correction factor has not been applied."

In line 350 we rephrased the statement to "Taking a potential offset between the two records into account".

L352-353: how about PCP?

Section added.

L365: fast dripping =>fast growth: not necessarily. Fast drip sites in cave may be characterized by undersaturated solution with respect to CaCO3, and may not lead to CaCO3 precipitation (but instead, corrosion of the earlier deposited carbonates)

See above. We have no idea about the original drip rate of the speleothem as the speleothem stopped growing about 2ka ago. However as we do not see evidence for corrosion on the

speleothem, we can safely assume that the the drip water always reached the top of the speolethem in a state of supersaturation with respect to the Ca2+ concentration. And as long as the drip water arrives at the drip site in a supersaturated condition, drip rate is a major driver for growth rate (see speleothem modelling studies; e.g., Dreybrodt and Scholz, 2011; Scholz et al., 2009)

L370: low drip rate may in some cases increases the CaCO3 precipitation rates, unless the number of drops/sec is very slow, and hence could lead in hiatus in deposition

We are sorry, but we cannot follow the argument provided here. Following cave modelling studies about growth rate systematics of speleothems it is always the case, that longer drip intervals lead to reduced speleothem growth rates (e.g., Dreybrodt and Scholz, 2011; Scholz et al., 2009) as long as a Ca2+ supersaturated solution impinges on the stalagmite top. We are not aware of the contrary from any study.

L378: Fractionation effects: please be precise, this title is quite vague

We are now more precise and termed the title „changes in fractionation effects"

L381: lighter molecules change…. Please rephrase, it reads awkward

changed to: "are preferentially transferred"

L382: Please delete : "this is valid for 13C and 14C isotopes". The statement does not bring any novelty

Done

L386: any figures for reference here?

No. This statement follows from the conventionally and generally applied concept of fractionation correction of radiocarbon measurements by d13C values (Stuiver and Pollach, 1977).

L388–390: the authors should elaborate on this with one or two sentences

If changes in fractionation occur, then this has some imprint on d13C and d18O. Both proxies should correlate over a short time interval. This was highlighted and explained in the sentence before. In the sentence in line 388-290 we only mention the method, with which we tested for this in a section wise and time resolving approach. We think that is already well explained.

L392: FigA7: this figure does not exist? Or did I miss it?

We are sorry. It should read Fig. S7. We improved this.

L398: long drip interval: I think if drip interval is long, this could enhance degassing/evaporation

We are not sure if we understand the reviewer with respect to the degassing argument. CO2 degassing is always the first process what happens (see. e.g., Dreybrodt and Scholz, 2011; Guo and Zhou, 2019). The degassing provides the means for the solution becoming supersaturated with respect to Ca2+, which finally allows CaCO3 precipitation. After the initial degassing, further CO2 degassing is only triggered by CaCO3 precipitation as for each CaCO3 molecule one CO2 molecule will be released.

Evaporation processes should be minimal as the cave has a high relative humidity close to 100%.

L403: which is approximately -8permil (and please add reference)

Added reference.

L408: ref. please?

Added reference.

L428-431: here the authors clearly state that their approach could bring light on understanding processes affecting C isotopes in the subsurface (one of the reasons I made a comment about the mismatch between the title and the content of the manuscript)

See above.

L442: contributed CO2 to CaCO3: this is awkward, please rephrase

Rephrased to "We propose that climatic conditions have changed such that this old OM pool in the karst, which is decoupled from the atmosphere, is being increasingly decomposed resulting in a

successively increasing $CO_2$ concentration in the karst. This $CO_2$ results in acidification in a comparable way as soil $CO_2$ enhancing carbonate dissolution and ultimately contributes to stalagmite $CaCO_3$. The isotopic C imprint, however, is significantly different to soil $CO_2$ causing the observed increase in dcf."

L463: why only CO2 (shouldn't be DIC that is more appropriate?)

Done

Figure 6 is very confusing to me; I am not fully sure what information to take from here?

What is the importance of elevation in F14C?

We improved the figure outline and use dcf now instead of F14C. In the text we describe, that it was very likely, that there was more vegetation than today during the early Holocene, as the timberline was higher, this allowed to produce more SOM, which become uncoupled from the atmosphere later on. As soon as the timberline decreases only a few grasses are expected to survive. Those contribute little CO2. We expect the majority of soil/karst CO to come from old organic material established during the early Holocene. Under such conditions we would expect to see an ageing component in the F14C or dcf. This observation is indicated by the dashed line.

L477: Re-writing suggestion:

"Results from this study allow to distinguish three intervals with different carbon dynamics.

(1) The interval before 8ka BP….

(2) The interval between 8–3.8ka BP (switch the number to follow the temporal logic)

(3) ……"

Done

We changed the numbers the older being mentioned first and the younger date second throughout the manuscript.

L479: potentially enhancing bedrock dissolution

Done

**Additional changes:**

We corrected the use of English language throughout the manuscript and corrected misspelt words.

L30: rephrased to: This novel approach can provide radiocarbon data at a spatial resolution similar to that of stable carbon [..]

L33: comma added

Line 44: "contributed" instead of "is contributing"

Line 45: "was" instead of "is"

Line 47: "hint" instead of "are hinting"

L98: added "(marine-derived)»

L106: rephrased to: "We investigate a stalagmite that grew in the high-alpine Spannagel cave system(47.080278°N, 11.671667°E, Tyrol, Austria; Spötl et al., 2004) by means of C isotope systematics."

L130: parenthesis removed

L137: "carbon" added

L140: "(compare Fig. 1)" removed

Section Results: "1. Radiocarbon" moved to SI (as suggested by reviewers) and content changed accordingly.

L301: "F14C of 0" replaced with "devoid of 14C".

L328-329: sentence rephrased by adding "speleothem" and "consequently".

L406: we added a more accurate value for the d13C of the bedrock: +2.5‰ instead of 0‰

L409-410: we adjusted our calculated values from "-12 and -11‰" to "-11.5 and -9‰"

---

## Referee Report (RR1)

**Review of the revised version of CP-2020-110 "Climatic variations during the Holocene inferred from radiocarbon and stable carbon isotopes in a high Alpine cave" by Welte et al.**

I would like to thank the authors for taking the time to address the comments and feedback from the reviewers. The manuscript has indeed improved, however, I feel that one fundamental notion related to the C dynamics needs to be considered, as I notice while reading the revised manuscript that the dissolved inorganic carbon (DIC) is likely to be confused with dissolved organic carbon (DOC). This did not occur to me, until I read the last paragraph of the discussion (L390), stating that "*the majority of DIC that contributed to the speleothem CaCO₃ had its origin in aged soil OM*". I don't think this is correct, DIC= Dissolved **Inorganic** Carbon, and inorganic carbon is different from organic carbon (See for example Lechleitner et al., 2019). The distinction between DIC and DOC may be the main source of misconception, and potentially a misleading interpretation of the d13C variability. Here, I suggest a re-consideration of this conclusion, and may be a careful revision of the discussion.

While re-reading the revised manuscript, I also feel that the three sections of the discussions are relatively disproportionate, with the middle section being the longest. I understand the reasons for the subdivisions (as it stands), however, the rationale for the division is not fully convincing, time-wise and data-wise (see the following paragraph). I'd suggest having ***a discussion section that pertains to Fig. 4 (before section 5.1).*** Any possible explanations of the climatic/environmental changes in either selected interval (i.e., before 8ka, after 3.8 ka, or the interval in between) could be grouped together in this section as a generalization. With all the assumptions and ideas grouped together here, it could be easier to later discuss the driving climatic changes for each selected interval, with more details given to the middle interval.

More about the subdivision of their records in 3 sections, the subtitle for the current sections 5.1, 5.2, and 5.3 is not parallel (hence again my suggestions above), and although the authors state that this subdivision is based on the dcf and d13C behavior, the timeframe rationale is not fully convincing.  To be clearer with what I meant: the interval before 8ka lasts only ~0.5ka and the interval after 3.8 ka lasts also ~ 1.4ka, however, the mid- interval is quite long (almost 4ka), and there are some intervals (e.g. between 5-6 ka), with relatively stable dcf and d13c, but only the jumps in the record are put in highlight. If the interpretation of the pre-8ka (very short time period) is also applied here (i.e., stable), I started to wonder, why the interval between 5 and 6 ka is also not considered stable? In fact, the 5-6ka interval appears to show less variability than the pre-8ka. I think that if the authors re-group all the discussion as I mentioned above, and then re-consider the subdivisions (approx. a millennial subdivision), the data could definitely make a very nice climatic and environmental history about the Spanagel's cave region.

There are some remarks raised by Reviewer #1 for previous manuscript (L 362 and L 395-427), which I would also like the authors to provide more clarification because the argument "gas exchange process" is a bit ambiguous. Clearly at L322 of the revised manuscript (and the corresponding paragraph, two different technical terms are used: **1.** Gas exchange process, and **2.** C isotope exchange. These are completely different mechanisms, and if I am not mistaken, the most common process during stalagmite formation is degassing (given that the percolating water from the soil and the epikarst above the cave is more saturated in $CO_2$ than the cave atmosphere). This is because the dripwater $pCO_2$ is in the process of equilibrating with the cave atmosphere pCO2. Otherwise, I am also confused with this gas exchange process. My other suggestions would be to discuss about $CO_2$ (de)hydration-(de)hydroxylation, if that is what the authors

really meant by this gas exchange process (e.g., McConnaughey, 1989; Usdowski et al., 1991), but otherwise, I am confused.

**In text-comments**

L19: instead of saying very high, just directly say the spatial resolution (in mm or in micrometer). It is better to be accurate

L21: please add microdrill for precision here. IRMS does not do the same job as LA-AMS. It needs more tools to extract the sample before it can analyze the samples

L30: variability in d13C with values ranging from -8 t +1 per mil and a generally lower dcf (the underlined texts are suggestion for rephrasing)

L 30-32: something is not correct in this sentence

why not only saying degassing (the 'gas exchange process' is perplexing and it sounds as if there are more gas in the cave. .but only cave air with $CO_2$)

L 36: the potential (please remove high, it's just making the word 'potential' weak and redundant)

L44: may be add Cheng et al., 2013 for more recent techniques?

L52: add coma before "decaying" and after "2013)"

L55: the analyses

L61: add space Fohlmeister

L67: to be technically correct, you could say "is expected to vary considerably". There is no expectation of a value to be complex, it is the process leading to the variations that is complex. Value and processes should not be confused.

L75: I think you should also add in this paragraph (i.e., mention quickly) the pyrite oxidation so that readers are aware of the possible cause of acid dissolution of the host carbonate. The detailed explanation about pyrite oxidation can remain where they are below, but it is a good idea to list them all at once in this paragraph.

L76: I think this sentence is lacking some info about rainfall (soil carbonic acid does not exist without water from rainfall)

L78: add the ref. Bergel et al. 2017, Noronha et al., .2015 after "(SOM)", and add relevant reference for the second source, i.e., after "through the soil", and delete the following sentence "*Recently, evidence was found for a potential additional C source stemming from $CO_2$ derived from the oxidation of "old" organic matter (OM) in the deep vadose zone (Bergel et al., 2017; Noronha et al.,2015)*". Then, see my comment above (L:75) to be included here.

L82: stable and radioactive instead of radiocarbon (otherwise, it is not parallel)

L83: Figure 1: In this figure, you show soil as a source of OM, do you refer here as the root respiration? If yes, please update the figure to accurately reflect the text, or update the text + figure to fully incorporate all the sources.

Some other instances about soil source are also discussed in the remaining parts of the manuscript (e.g. the sentences after L97), and could be helpful if presented here?

L88: about this 'reservoir effect', is there a difference between the source/factor as you discussed above (OM decay, root respiration, and pyrite oxidation), or does this reservoir effect only apply to OM?

L102: please add a reference after "50%"

Table 1: some key ref. should be added for Table 1

L127: my knowledge about the mineralogy of gneiss is as follow: quartz, Feldspar, hornblende, and biotite. Here the authors do not really specify if this is a pyrite-bearing gneiss or if the pyrite exists as a vein

somewhere in the outcrops. Some explanations are needed here. Otherwise, a relevant reference after "gneiss" (e.g., some papers reporting on the geology of the study region)

L133: with an average growth rate of 25μm/a based on 9 U/Th ages

L134: About hiatus: did the authors have any microscopic evidence for this (any figure for reference?) Also coeval existence of other stalagmites from the same cave does not explain the absence of micro-hiatus in a sample. I suspect that Rev. 1 is correct (if the authors do no have any petrographic evidence to support their claim, I'd suggest responding carefully to Rev. 1 and considering this in the manuscript)

L139: may be "interrupted" instead of "affected" is more grammatically correct here

L155: better to indicate the lab where the analyses were performed as in L147

L172: indicate the section number (as the subsections in your manuscript are numbered)

Figure 2: is it possible to add in this figure the location of the age trench, please.

The handwriting is a bit confusing, and if looking at Figure 3, the dft goes beyond 1400, where exactly are these trenches located?

Also, it seems that the figure for the bottom pieces have been removed? Should the word "top and bottom piece" be updated in the caption?

L191: Please rephrase as scan ref. T1, T3, and B3 so that readers don't get confused with two pieces and the scanning method.

L192: "From the 14C profile" -- I think you mean "Using the dft from the 14C scans"?

L194: do you mean 1502 instead of 1402? (see previous comment by Reviewer 1, L243)

L195: some inconsistent labelling, please correct (there are many instances in the manuscript that the authors need to check and correct. If in the figure they use "a", they should keep that labelling in the text and not change it to "A")

L198-203: give the estimated age for each interval

L205-206: This discussion will be divided in three sections, which are based on the speleothems d13C and dcf characteristics (this is a suggestion for re-wording, however, please consider my general comments above)

L211: I think it is better to provide the carbonate d13C values under C3 vegetation rather than this -25

L215: greater instead of larger (dcf values do not represent size, thus, they can't be large or small)

L216-217: the presence of two "which" complicates this sentence, please simplify (either by removing the second explanation with "which" or by splitting in two sentences

L222: "falls coincides"—please revise

L229: Figure 3: Please indicate in this figure the three subdivision you mention in the discussion (may be with a horizontal bar, and annotation I, II, and III)—also, please revise the title of the subsection of your discussion. And again, with the figure labelling and the use of capital or lowercase character (a vs A, b vs B), please be consistent with both the figures and the text.

L235: of (instead of ofs)

L236: to be specific, may be better with 3c?

L241: too wordy, "…processes must be considered"

Additional general comment about this paragraph: In some part, I may agree with Referee 1 with some of the comments. The high dcf values at 5-6ka (which is closer in amplitude to the interval before 8ka. In fact, the section before 8 and after 3.8 ka are quite short compared to the mid-interval. I think the rationale in the subdivision is not fully convincing. The authors just left the middle interval as is and may seem to overgeneralize it rather than classifying it more appropriately based on the degree of correlation between d13C and 14C.

L249: Now, as I re-read the discussion, this section about the 5-6ka should be treated separately, and not shadowed under section II (see again my general comments at the beginning of this review)

L253: Do you mean proposing two hypotheses? (I don't think you are testing a hypothesis here, you're proposing xx and based on your data, you make assumptions on which factors/drivers could best explain your case)

L257: the use of soil-derived CO2 needs to be specified because it is unclear if this is soil-respired from plant roots or from decaying om in the soil

L267: why DCF in capitals?

L272: Considering the potential offset between…

L273: use the section number..

For the positive correlation, can you please provide the r2 and the p-values?

L279-283: please rephrase and write clearer

L294: Please indicate at the end of the sentence the corresponding growth rate for the older section (a good reminder for the readers)

L297-300: if the drip water pCO2 is low, as hypothesized by the authors, the $CO_2$ gradient between the drip and cave is small, hence, degassing is less, and thus should not lead to a significant increase in d13C (via degassing)

L303: do you mean "isotope fractionation"? or something else?

L318 (about PCP): I think it would be nice if in Figure 3, d13C and d18O are plotted as a time series together (such as the d13C-dcf pair), it is currently difficult to see that in the current figure, and then to assess the PCP

L320: "While this would not have an effect on 14C.." Why not? Any reference to support this claim?

L322: I think the term "gas exchange" is relatively incorrect, because the drip water is mainly composed of DIC, so it could be more technically correct to say DIC exchange with cave atmosphere. Gas exchange is more likely believed to occur in open system (e.g., soil pore $CO_2$ exchange with atmospheric $CO_2$). Please be careful with inaccurate terminology.

L324: I think this terminology about C-isotope exchange is more accurate

L325: this term "are long" is vague (and one of the reasons for my comment in the previous version). Please specify the time for the drip intervals (how many drips per second?)

L332: why two biogenic sources and providing only one value, and what are they?

L341: reference please after 5°C

L352-355: this does not belong here and does not provide any conclusions about the large changes between 5-6ka. Please revise (following the guidance above)

L376-…: although this is a great story about the landscape change in the region, the d13C seems to contradict with the statement (look that the d13C values are very high, and these are not typical for trees nor C3 plants

L393: With the novel LA-AMS…

L397: This interval before 8ka is only covering a period of <1ka? (which is comparable to the interval of 5–6k where both d13C and dfc also appears stable, but higher). As noted above, I think the choice of time interval subdivision is not fully convincing, leaving the longest part of the record (between 8 and 3.8ka) under debate. Please consider the general comments above.

L401: I think some of the arguments presented in this paragraph still support PCP. Because of the reduced meteoric precipitation, the water percolating down to the cave is not sufficient to reach the cave itself. Instead, it precipitated $CaCO_3$ in the epikarst prior to reaching the top of the stalagmite, hence PCP.

L405: this is the reason why pyrite oxidation should be shown in Figure 1
L424: please revise (FTIR no longer belongs to the manuscript)
L425: helped improving this manuscript.

**Supplementary:**
Figure 5: texts are too small, please revise
Figure S10: It would be better if the width of this figure is expanded to fill the entire width of the page and reduce the height by 2/3.
References: Please update reference style to match Copernicus's

**References:**
Cheng, H., Edwards, R.L., Shen, C.C., Polyak, V.J., Asmerom, Y., Woodhead, J., Hellstrom, J., Wang, Y., Kong, X., Spötl, C. and Wang, X., 2013. Improvements in 230Th dating, $^{230}$Th and $^{234}$U half-life values, and U–Th isotopic measurements by multi-collector inductively coupled plasma mass spectrometry. Earth and Planetary Science Letters, 371, 82–91.

Lechleitner, F., Lang, S., Haghipour, N., McIntyre, C., Baldini, J., Prufer, K., & Eglinton, T. (2019). Towards Organic Carbon Isotope Records from Stalagmites: Coupled δ13C and 14C Analysis Using Wet Chemical Oxidation. Radiocarbon, 61(3), 749-764. doi:10.1017/RDC.2019.35

McConnaughey T. (1989) 13C and 18O isotopic disequilibrium in biological carbonates: I. Patterns. Geochim. Cosmochim. Acta 53, 151–162.

Usdowski E., Michaelis J., Bottcher M. E. and Hoefs J. (1991) Factors for the oxygen isotope equilibrium fractionation between aqueous and gaseous CO2, carbonic-acid, bicarbonate, carbonate, and water (19 °C). Z. Phys. Chem. 170, 23 7–249.

---

## Author Response (AR2)

RESPONSE TO THE REVIEWER'S COMMENTS:

I would like to thank the authors for taking the time to address the comments and feedback from the reviewers. The manuscript has indeed improved, however, I feel that one fundamental notion related to the C dynamics needs to be considered, as I notice while reading the revised manuscript that the dissolved inorganic carbon (DIC) is likely to be confused with dissolved organic carbon (DOC). This did not occur to me, until I read the last paragraph of the discussion (L390), stating that "the majority of DIC that contributed to the speleothem CaCO3 had its origin in aged soil OM". I don't think this is correct, DIC= Dissolved Inorganic Carbon, and inorganic carbon is different from organic carbon (See for example Lechleitner et al., 2019). The distinction between DIC and DOC may be the main source of misconception, and potentially a misleading interpretation of the d13C variability. Here, I suggest a re-consideration of this conclusion, and may be a careful revision of the discussion.

We think that the reviewer misunderstood our methodology (LA-AMS): we can only detect the DIC in the stalagmite and not DOC with our method. The DOC is washed down into the karst system where it is decomposed likely by microbial activity, thus, contributing to the DIC in the seepage water. This signal then contributes to the C in the speleothem CaCO3. This is explained in lines 79 – 81.

While re-reading the revised manuscript, I also feel that the three sections of the discussions are relatively disproportionate, with the middle section being the longest. I understand the reasons for the subdivisions (as it stands), however, the rationale for the division is not fully convincing, time-wise and data-wise (see the following paragraph). I'd suggest having a discussion section that pertains to Fig. 4 (before section 5.1). Any possible explanations of the climatic/environmental changes in either selected interval (i.e., before 8ka, after 3.8 ka, or the interval in between) could be grouped together in this section as a generalization. With all the assumptions and ideas grouped together here, it could be easier to later discuss the driving climatic changes for each selected interval, with more details given to the middle interval.

More about the subdivision of their records in 3 sections, the subtitle for the current sections 5.1, 5.2, and 5.3 is not parallel (hence again my suggestions above), and although the authors state that this subdivision is based on the dcf and d13C behavior, the timeframe rationale is not fully convincing. To be clearer with what I meant: the interval before 8ka lasts only ~0.5ka and the interval after 3.8 ka lasts also ~ 1.4ka, however, the mid- interval is quite long (almost 4ka), and there are some intervals (e.g. between 5-6 ka), with relatively stable dcf and d13c, but only the jumps in the record are put in highlight. If the interpretation of the pre-8ka (very short time period) is also applied here (i.e., stable), I started to wonder, why the interval between 5 and 6 ka is also not considered stable? In fact, the 5-6ka interval appears to show less variability than the pre-8ka. I think that if the authors re-group all the discussion as I mentioned above, and then reconsider the subdivisions (approx. a millennial subdivision), the data could definitely make a very nice climatic and environmental history about the Spanagel's cave region.

Here, we are answering both of the previous paragraphs combined. We think that a discussion on a millennial scale is not possible based on our data and still think that the three sections we chose are appropriate. We chose the d13C signature as the basis for the different sections as it exhibits two relatively stable intervals (i.e., > 8 ka and <3.5 ka). The 14C data assist in the interpretation of the relevant processes in each period. The longest section in the middle (3.5 – 8 ka) shows strong fluctuations in d13C.
To make our approach clearer, we added the following sentence to L 205/206: " [..], i.e., two periods of comparably stable $\delta^{13}$C before 3.5 ka BP and after 8 ka BP and the interval in between with large and rapid fluctuations."

There are some remarks raised by Reviewer #1 for previous manuscript (L 362 and L 395-427), which I would also like the authors to provide more clarification because the argument "gas exchange process" is a bit ambiguous. Clearly at L322 of the revised manuscript (and the corresponding paragraph, two different technical terms are used: 1. Gas exchange process, and 2. C isotope exchange. These are completely different mechanisms, and if I am not mistaken, the most common process during stalagmite formation is degassing (given that the percolating water from the soil and the epikarst above the cave is more saturated in CO2 than the cave atmosphere). This is because the dripwater $pCO_2$ is in the process of equilibrating with the cave atmosphere $pCO_2$. Otherwise, I am also confused with this gas exchange process. My other suggestions would be to discuss about CO2 (de)hydration-(de)hydroxylation, if that is what the authors really meant by this gas exchange process (e.g., McConnaughey, 1989; Usdowski et al., 1991), but otherwise, I am confused.

We agree with the reviewer and are now more precise about the terminology. Instead of "gas exchange" we will use "C exchange between cave air $CO_2$ and DIC in drip water" throughout the manuscript.
We do discuss "degassing" as an important process in the manuscript, however, the combination of d13C, d18O and dcf indicates that this process is not enough to explain our observations as we would expect a similar behavior in d18O.

**In text-comments**
L19: instead of saying very high, just directly say the spatial resolution (in mm or in micrometer). It is better to be accurate
We added: "at spatial resolution down to 100 μm"
L21: please add microdrill for precision here. IRMS does not do the same job as LA-AMS. It needs more tools to extract the sample before it can analyze the samples
We added: "of micromilled samples"
L30: variability in d13C with values ranging from -8 t +1 per mil and a generally lower dcf (the underlined texts are suggestion for rephrasing)
Changed accordingly.
L 30-32: something is not correct in this sentence why not only saying degassing (the 'gas exchange process' is perplexing and it sounds as if there are more gas in the cave. .but only cave air with $CO_2$)
As stated above, we changed the expression "gas exchange" to "C exchange between cave air $CO_2$ and dissolved inorganic carbon in drip water" throughout the manuscript.
L 36: the potential (please remove high, it's just making the word 'potential' weak and redundant)
Done.
L44: may be add Cheng et al., 2013 for more recent techniques?
Done.
L52: add coma before "decaying" and after "2013)"
Done.
L55: the analyses
Done.
L61: add space Fohlmeister
Done.
L67: to be technically correct, you could say "is expected to vary considerably". There is no expectation of a value to be complex, it is the process leading to the variations that is complex. Value and processes should not be confused.
Changed accordingly.
L75: I think you should also add in this paragraph (i.e., mention quickly) the pyrite oxidation so that readers are aware of the possible cause of acid dissolution of the host carbonate. The detailed explanation about pyrite oxidation can remain where they are below, but it is a good idea to list them all at once in this paragraph.
L81: added "In some karst systems, the oxidation of pyrite has shown to contribute to the acidification of the seepage water and hence to speleothem formation (e.g., Spötl et al., 2016)»

L76: I think this sentence is lacking some info about rainfall (soil carbonic acid does not exist without water from rainfall)

Added "In most karst systems, dissolution of the carbonate host rock is driven by soil-derived carbonic acid forming in meteoric precipitation seeping through the soil."

L78: add the ref. Bergel et al. 2017, Noronha et al., .2015 after "(SOM)", and add relevant reference for the second source, i.e., after "through the soil", and delete the following sentence "*Recently, evidence was found for a potential additional C source stemming from CO₂ derived from the oxidation of "old" organic matter (OM) in the deep vadose zone (Bergel et al., 2017; Noronha et al.,2015)*". Then, see my comment above (L:75) to be included here.

Reference added for root respiration (Cerling 1984) and (Trumbore 2000) for SOM degradation.

We did not change the structure of this paragraph, because we introduce three different processes that all lead to the contribution of organic C in speleothem carbonate: 1. CO2 stemming from the degradation of soil organic matter through microbial activity, 2. root respiration and 3. degradation of old organic matter deep down in the karst.

L82: stable and radioactive instead of radiocarbon (otherwise, it is not parallel)

Done.

L83: Figure 1: In this figure, you show soil as a source of OM, do you refer here as the root respiration? If yes, please update the figure to accurately reflect the text, or update the text + figure to fully incorporate all the sources. Some other instances about soil source are also discussed in the remaining parts of the manuscript (e.g. the sentences after L97), and could be helpful if presented here?

Done.

L88: about this 'reservoir effect', is there a difference between the source/factor as you discussed above (OM decay, root respiration, and pyrite oxidation), or does this reservoir effect only apply to OM?

The dcf includes everything that causes a deviation of the 14C content in the stalagmite CaCO3 relative to the atmosphere at the time of stalagmite formation. This is explained in section 2 of the manuscript.

L102: please add a reference after "50%"

We added «Hendy 1971»

Table 1: some key ref. should be added for Table 1

We added to the caption: "This table is a compilation of data from Fohlmeister 2011b, Spötl et al. 2016, and Therre et al. 2020."

L127: my knowledge about the mineralogy of gneiss is as follow: quartz, Feldspar, hornblende, and biotite. Here the authors do not really specify if this is a pyrite-bearing gneiss or if the pyrite exists as a vein somewhere in the outcrops. Some explanations are needed here. Otherwise, a relevant reference after "gneiss" (e.g., some papers reporting on the geology of the study region)

Pyrite occurs as accessory mineral in this gneiss. We added a reference (Spötl et al., 2004) in L 62.

L133: with an average growth rate of 25µm/a based on 9 U/Th ages

Changed accordingly.

L134: About hiatus: did the authors have any microscopic evidence for this (any figure for reference?) Also coeval existence of other stalagmites from the same cave does not explain the absence of micro-hiatus in a sample. I suspect that Rev. 1 is correct (if the authors do no have any petrographic evidence to support their claim, I'd suggest responding carefully to Rev. 1 and considering this in the manuscript)

May be the reviewer misunderstood something: The depth-age model shows a gradual age progression (Fig. 3C) and no evidence of growth rate changes. And we added in the previous manuscript version that also the macro- and microscopic inspection of the polished slab shows no petrographic evidence of a hiatus (previous version l 134, now in l 141-145).

L139: may be "interrupted" instead of "affected" is more grammatically correct here

Changed accordingly.

L155: better to indicate the lab where the analyses were performed as in L147

Added: "at the Laboratory of Ion Beam Physics, ETH Zurich, Switzerland,"

L172: indicate the section number (as the subsections in your manuscript are numbered)

Changed accordingly.

Figure 2: is it possible to add in this figure the location of the age trench, please.

Done. Caption added: "Roman numbers and dashed black lines mark the three sections discussed separately."

The handwriting is a bit confusing, and if looking at Figure 3, the dft goes beyond 1400, where exactly are these trenches located?

Also, it seems that the figure for the bottom pieces have been removed? Should the word "top and bottom piece" be updated in the caption?

"Top and bottom piece" is removed.

L191: Please rephrase as scan ref. T1, T3, and B3 so that readers don't get confused with two pieces and the scanning method.

Done.

L192: "From the 14C profile" -- I think you mean "Using the dft from the 14C scans"?

Rephrased to: "Using the 14C profile of the speleothem, the StalAge (Scholz and Hoffmann, 2011) age-depth model applied to previously published U-Th data (Fohlmeister et al., 2013) and the known 14C content of the atmosphere during the Holocene, …"

L194: do you mean 1502 instead of 1402? (see previous comment by Reviewer 1, L243)

No, this is the correct and final number. It was a mistake.

L195: some inconsistent labelling, please correct (there are many instances in the manuscript that the authors need to check and correct. If in the figure they use "a", they should keep that labelling in the text and not change it to "A")

Changed accordingly.

L198-203: give the estimated age for each interval

Changed to: "between 30 and 130 mm (ca. 4.0 - 8.1 ka BP). Layers exhibiting a comparably stable δ13C occur at the top and bottom of SPA 127, specifically ranging from 4 to 25 mm (ca. 2.6 and 3.7 ka BP) and from 130 to 144 mm (ca. 8.1 – 8.4 ka BP)."

L205-206: This discussion will be divided in three sections, which are based on the speleothems d13C and dcf characteristics (this is a suggestion for re-wording, however, please consider my general comments above)

Done and we also considered the other comments.

L211: I think it is better to provide the carbonate d13C values under C3 vegetation rather than this -25

We rewrote this sentence to make it clearer: "The relatively low δ13C value of -5‰ actually contradicts this [..]"

L215: greater instead of larger (dcf values do not represent size, thus, they can't be large or small)

Done.

L216-217: the presence of two "which" complicates this sentence, please simplify (either by removing the second explanation with "which" or by splitting in two sentences

Rephrased to: Thus, in addition to the atmospheric radiocarbon contribution from living vegetation, an "old" OM source, which respires radiocarbon-depleted CO2, is required to explain the depleted δ13C values and elevated dcf.

L222: "falls coincides"—please revise

Removed "falls"

L229: Figure 3: Please indicate in this figure the three subdivision you mention in the discussion (may be with a horizontal bar, and annotation I, II, and III)—also, please revise the title of the subsection of your discussion. And again, with the figure labelling and the use of capital or lowercase character (a vs A, b vs B), please be consistent with both the figures and the text.

Done.

L235: of (instead of ofs)

Done.

L236: to be specific, may be better with 3c?

Done.

L241: too wordy, "…processes must be considered"

Done.

Additional general comment about this paragraph: In some part, I may agree with Referee 1 with some of the comments. The high dcf values at 5-6ka (which is closer in amplitude to the interval before 8ka. In fact, the section before 8 and after 3.8 ka are quite short compared to the mid-interval. I think the rationale in the subdivision is not fully convincing. The authors just left the middle interval as is and may seem to overgeneralize it rather than classifying it more appropriately based on the degree of correlation between d13C and 14C.

As stated above, our choice of division is based on the d13C signature. The 14C signal assists in the interpretation of the d13C signal, but a large proportion of the dcf remains elusive. This includes the prominent dcf feature around 5 – 6 ka.

L249: Now, as I re-read the discussion, this section about the 5-6ka should be treated separately, and not shadowed under section II (see again my general comments at the beginning of this review)

We are not shadowing the section between 5 – 6 ka but discuss it within the according subdivision (Section 5.2; L 249 – 254).

L253: Do you mean proposing two hypotheses? (I don't think you are testing a hypothesis here, you're proposing xx and based on your data, you make assumptions on which factors/drivers could best explain your case)

Changed accordingly.

L257: the use of soil-derived CO2 needs to be specified because it is unclear if this is soil-respired from plant roots or from decaying om in the soil

We added "(from root respiration and microbial decomposition of SOM)"

L267: why DCF in capitals?

Changed accordingly

L272: Considering the potential offset between…

Changed accordingly

L273: use the section number..

Changed accordingly

For the positive correlation, can you please provide the r2 and the p-values?

We changed this to «a high degree of similarity», because of the difficulty in aligning the two data sets, the correlation cannot be mathematically calculated, but only qualitatively observed (L273 and 280);

L279-283: please rephrase and write clearer

Rephrased to: "Changes between open and closed carbonate dissolution regimes are expected to result in a correlation between the dcf and d13C, which is observed in SPA 127. However, the magnitude of d13C variations is too large to be explained by a change of dissolution regime even when considering the extreme switch from completely open to completely closed."

L294: Please indicate at the end of the sentence the corresponding growth rate for the older section (a good reminder for the readers)

Done.

L297-300: if the drip water pCO2 is low, as hypothesized by the authors, the $CO_2$ gradient between the drip and cave is small, hence, degassing is less, and thus should not lead to a significant increase in d13C (via degassing)

We agree with the reviewer (see Guo 2020 GCA) and removed the term "fractionation" from line 299.

L303: do you mean "isotope fractionation"? or something else?

Yes, changed accordingly.

L318 (about PCP): I think it would be nice if in Figure 3, d13C and d18O are plotted as a time series together (such as the d13C-dcf pair), it is currently difficult to see that in the current figure, and then to assess the PCP

Changed accordingly

L320: "While this would not have an effect on 14C." Why not? Any reference to support this claim?

When measuring 14C through AMS, a fractionation correction is applied conventionally for in the radiocarbon community. Thus, all samples are reported as if they had a d13C value of -25 per mill and all fractionation, which occur to 14C, would have been corrected for. We added a sentence to the materials & methods section (L 175): "and conventionally a fractionation correction to a δ13C of -25‰ was applied (Stuiver and Polach, 1977)."

L322: I think the term "gas exchange" is relatively incorrect, because the drip water is mainly composed o DIC, so it could be more technically correct to say DIC exchange with cave atmosphere. Gas exchange is more likely believed to occur in open system (e.g., soil pore $CO_2$ exchange with atmospheric $CO_2$). Please be careful with inaccurate terminology.

Changed throughout the manuscript (see above).

L324: I think this terminology about C-isotope exchange is more accurate

Changed throughout the manuscript (see above).

L325: this term "are long" is vague (and one of the reasons for my comment in the previous version). Please specify the time for the drip intervals (how many drips per second?)

Rewritten as: "meaning that 95% of the $CaCO_3$ precipitated either via PCP or on the stalagmite (Guo, 2020), "

L332: why two biogenic sources and providing only one value, and what are they?

Because the three biogenic sources have the same d13C value but potentially different 14C values (see Fig. 1 and section 2); see also above.

L341: reference please after 5°C

We added Mook et al. (1974).

L352-355: this does not belong here and does not provide any conclusions about the large changes between 5-6ka. Please revise (following the guidance above)

We agree with the reviewer and removed this section.

L376-…: although this is a great story about the landscape change in the region, the d13C seems to contradict with the statement (look that the d13C values are very high, and these are not typical for trees nor C3 plants

The d13C is the stalagmite is the result of mixing of different sources and numerous processes. This has been discussed in detail throughout the manuscript (see Table 1, Fig. 1 and section 2).

L393: With the novel LA-AMS…

Done.

L397: This interval before 8ka is only covering a period of <1ka? (which is comparable to the interval of 5–6k where both d13C and dfc also appears stable, but higher). As noted above, I think the choice of time interval subdivision is not fully convincing, leaving the longest part of the record (between 8 and 3.8ka) under debate. Please consider the general comments above.

See above.

L401: I think some of the arguments presented in this paragraph still support PCP. Because of the reduced meteoric precipitation, the water percolating down to the cave is not sufficient to reach the cave itself. Instead, it precipitated $CaCO_3$ in the epikarst prior to reaching the top of the stalagmite, hence PCP.

We don't understand this comment, because the stalagmite will not grow without water.

L405: this is the reason why pyrite oxidation should be shown in Figure 1

Ok.

L424: please revise (FTIR no longer belongs to the manuscript)

FTIR is still part of the manuscript, even if it is now only in the supplement. Therefore, we prefer to leave this acknowledgement sentence as is.

L425: helped improving this manuscript.

Done

**Supplementary:**
Figure 5: texts are too small, please revise

We agree that this would be nicer but it is due to logistical reasons not possible.

Figure S10: It would be better if the width of this figure is expanded to fill the entire width of the page and reduce the height by 2/3.

We prefer not to change the Fig.

References: Please update reference style to match Copernicus's

Done.

**Additional changes:**
Title: we added "in speleothems" to be more precise
L21: removed "(IRMS)
Caption Figure 1: spelt out OC (organic carbon)
Caption Figure 2: removed the explanation in the parenthesis (it is confusing and doesn't add relevant information)

L194: we added what the abbreviation "SG" means (Savitzky-Golay)